# Recession analysis revisited: impacts of climate on parameter estimation

Elizabeth R. Jachens[1], David E. Rupp[2], Clément Roques[3], John S. Selker[1]

[1]Department of Biological and Ecologic Engineering, Oregon State University, Corvallis, OR, 97330, USA

[2]Oregon Climate Change Research Institute, College of Earth, Oceanic, and Atmospheric Sciences, Oregon State University, Corvallis, OR, 97330, USA

[3]Department of Earth Sciences, ETH Zurich, 8092 Zürich, Switzerland

*Correspondence to*: Elizabeth R. Jachens (erjachens@gmail.com)

**Abstract.** Recession analysis is a classical method in hydrology to assess watersheds' hydrological properties by means of the receding limb of a hydrograph, frequently expressed as the rate of change in discharge ($-dQ/dt$) against discharge ($Q$). This relationship is often assumed to take the form of a power law $-dQ/dt=aQ^b$ where $a$ and $b$ are recession parameters. Recent studies have highlighted major differences in the estimation of the recession parameters depending on the method, casting doubt on our ability to properly evaluate and compare hydrological properties across watersheds based on $-dQ/dt$ vs. $Q$

recession analysis. This study shows that estimation based on collective recessions as an average watershed response is strongly affected by the distributions of event inter-arrival time, magnitudes, and antecedent conditions, implying that the resulting recession parameters do not represent watershed properties as much as they represent the climate. The main outcome from this work highlights that proper evaluation of watershed properties is only ensured by considering independently individual recession events. While average properties can be assessed by considering the average (or median) values of $a$ and $b$, their

variabilities provide critical insight into the sensitivity of a watershed to the initial conditions involved prior to each recharge event.

## 1 Introduction

Accurate representations of watershed-scale hydrological processes are urgent in a global and anthropogenic change perspective. Streamflow recession analysis has been routinely used for about half a century to assess watershed properties

(e.g., Brutsaert and Nieber, 1977; Kirchner, 2009; Mcmillan et al., 2014) and more recently their vulnerability to climatic and anthropogenic factors (Berghuijs et al., 2016; Brooks et al., 2015; Buttle, 2018; Fan et al., 2019). Recession analysis is commonly done by plotting the time rate of change in discharge $-dQ/dt$ vs. discharge $Q$ with bilogarithmic axes. Theory for

an idealized single aquifer predicts a power law relationship with parameters $a$ and $b$ (Brutsaert and Nieber, 1977; Rupp and Selker, 2005).

$$-dQ/dt = aQ^b \,, \tag{1}$$

However, it has long been recognized that the accuracy in the estimation of those parameters is highly sensitive to the methods used (Chen et al., 2018; Dralle et al., 2017; Roques et al., 2017; Rupp and Selker, 2006a; Santos et al., 2019; Stoelzle et al., 2013).

Two categories of parameter estimation methods are based on: 1) the aggregation of all observations in *-dQ/dt* vs. *Q* space, hereafter referred to as the "point cloud", to describe the average watershed behavior; and 2) the identification of individual recession events in *-dQ/dt vs. Q* space to look at the variability of a watershed's response. There is a long history of recession analysis parameter estimation using the point cloud beginning with Brutsaert and Nieber (1977) and it remains common (e.g., Buttle, 2018; Liu et al., 2016; Meriö et al., 2019; Ploum et al., 2019; Sánchez-Murillo et al., 2015; Stewart, 2015; Vannier et al., 2014; Yeh and Huang, 2019). In recent literature there has been a shift toward using individual recessions to estimate recession parameters (Basso et al., 2015; Karlsen et al., 2018; Roques et al., 2017) and Santos et al. (2019) go as far as to question the validity of point cloud estimation methods.

When Brutseart and Nieber (1977) first proposed their recession analysis method, aquifer recession behavior was identified by fitting a lower envelope to the point cloud, thus assuming small values of *-dQ/dt* for a given *Q* represent aquifer discharge flow and anything larger has contributions from faster pathways such as overland flow. This lower-envelope method of estimating recession analysis parameters was shown to be highly subject to artifacts arising from measurement noise and recording precision (Rupp and Selker, 2006a; Troch et al., 1993) and improvements to fitting a lower envelope have been proposed (Stoelzle et al., 2013; Thomas et al., 2015). An alternative fitting method wherein *b* was estimated as the best linear fit to the point cloud was introduced by Vogel and Kroll (1992) as the central tendency. The central tendency method was adapted by Kirchner (2009) to address the undue weight of highly uncertain extreme points. Kirchner (2009) also suggested fitting a polynomial function to averages within bins of the cloud data. All of these point cloud fitting approaches fundamentally treat each computation of *-dQ/dt* and *Q* as reflecting a single average underlying curve, with deviations from a single curve effectively treated as noise. In other studies, data have been subset by season or month (e.g., Szilagyi et al., 2007; Thomas et al., 2015) to examine seasonal variations in the recession characteristics with the subsets still treated to point cloud analyses.

In contrast, the variability in watershed response to individual recharge events can be depicted by fitting recession parameters to individual recession events. Several authors have observed that individual recessions had greater *b* than did the point cloud (Biswal and Marani, 2010; Mcmillan et al., 2011; 2014; Mutzner et al., 2013; Shaw and Riha, 2012); a larger value of *b*

indicates a time rate of decline that decreases more quickly with decreasing streamflow. Consistent with these studies, we have also observed individual recessions that have a larger $b$ than the point cloud fit across watersheds in the Oregon Cascades. As an example, we present in Fig. 1 recessions for Lookout Creek, Oregon, USA, using daily discharge data ($m^3$/s) from 1949 to 2016 (station USGS# 14161500) (Johnson and Rothacher, 2019; obtained from USGS, 2019). In the 66 years of data presented,

a total of 1309 recession events are identified with an average of 19 events per year. It is clear that values of $b$ for individual recession events tend to be larger than $b$ for the point cloud, particularly at lower discharges. In this example, individual event selection criteria include recessions lasting longer than 5 days, starting 1 day after the peak to exclude the influence of overland flow, and ending at the next precipitation event, following other studies (Biswal and Marani, 2010; Shaw and Riha, 2012). The $b$ parameter estimated using point cloud analysis (binning average method) is smaller ($b = 1.5$) compared to the median of $b$

values from the individual recessions ($b = 2.8$ with 50% of individual recessions taking values from 2.0 to 4.7; See colorbar of Fig. 1). The frequency distribution of the $b$ parameter from the individual recessions is skewed right and roughly lognormal which suggests that $b$ from the point cloud does not represent an average or 'master' recession behavior.

[Insert Fig. 1]

For a given discharge range in Fig. 1, there appear to be multiple individual recessions with similar values of $b$ that are

horizontally offset, implying a common $b$ but a variable $a$. The offset of individual recession events suggests that antecedent conditions may be influencing the location of the recession curves (e.g., Rupp et al., 2009), consistent with various theoretical definitions of $a$ that include the aquifer saturated thickness at the onset of the recession as a parameter (Rupp and Selker, 2006b). Many authors have associated the pattern of shifted individual recessions with seasonality (Bart and Hope, 2014; Dralle et al., 2015; Karlsen et al., 2018; McMillan et al., 2011; Shaw and Riha, 2012; Tashie et al., 2019). Generally, authors

describe a generally sinusoidal relationship with larger $a$ values associated with summer months (Dralle et al., 2015; Shaw and Riha, 2012) and a weaker seasonal relationship for values of $b$ (Karlsen et al., 2018; Tashie et al., 2019). Seasonality associated with meteorological conditions may well be used as a predictor of $a$ or $b$, but seasonality alone fails to address the underlying climatic conditions that control streamflow recession. Instead of describing the variability between events based on seasonality as a proxy, parameter estimation should focus on antecedent and meteorological conditions that control streamflow recession

in order to form a more comprehensive physically-based understanding of recession parameters (e.g., Bart and Hope, 2014; Karlsen et al., 2018).

This paper explores the source of the offset ($\ln(a)$) and slope ($b$) on individual recessions. Using a time-series of synthetic hydrographs with known parameters, we compare different methods for estimating the recession analysis parameters and the sensitivity to the method on the frequency and magnitude of events that make up the hydrograph. We are particularly concerned

with how individual recessions collectively create the emergent point cloud and seek to describe how recession parameter estimation of the point cloud is affected by the distribution of individual recessions.

## 2 Methods

This section presents methods for: 1) the definition of three synthetic hydrographs, 2) description of recession extraction from the hydrograph, and 3) the comparisons between four fitting methods for parameter estimation applied to a discharge time series for Lookout Creek.

5 **2.1 Synthetic Hydrograph Methods**

This paper makes use of synthetic hydrographs to explore factors that change $b$ for individual recession events as well as the inter-arrival times of individual events that create the point cloud. Our synthetic hydrographs are created by defining individual recession events and stitching them together to create a long time series. Synthetic hydrographs were chosen for this study because each individual recession can be definitively identified as the characteristics are known which is unrealistic when 10 considering real watersheds. Furthermore, the synthetic hydrographs can be specified to directly compare different climatic controls without the confounding variables traditionally associated with real watersheds. For these purposes, the specifications of the synthetic hydrographs were chosen to explore the effects of the magnitudes and frequency of recharge events on the recession analysis parameters from collective vs. individual recessions.

The falling limb of the hydrograph is assumed to follow a power law following Eq. (2) (Dewandel et al., 2003; Drogue, 1972; 15 Rupp and Woods, 2008):

$$Q(t) = Q_o \left( \frac{t}{\tau} + 1 \right)^{-w}, \tag{2}$$

where $Q$ is the discharge, $Q_o$ the initial discharge prior recession at $t$=0, $t$ is the time in days since the recession started, $\tau$ is a characteristic timescale, and $w$ is the dimensionless power-law decay exponent. Eq. (2) can be expressed as Eq. (1) with $a = w/(\tau Q_o^{1/w})$ and $b = (1+w)/w$.

20 Holding $\tau$ constant and varying the initial condition $Q_o$ results in a hysteretic $-dQ/dt$ vs. $Q$ relationship, in contrast to a constant $a$ value which produces a single non-hysteric relationship. Defining $a$ as a function of initial conditions has both theoretical (e.g., Rupp and Selker, 2006b) and empirical (e.g., Bart and Hope, 2014) support. The constancy of $\tau$ is not well established, but we assume it is constant for the scenarios examined here. Consequentially, a constant $\tau$ results in variable value for $a$ that is inversely proportional to the initial discharge. An inverse relationship is consistent with theoretical expectations for non- 25 linear aquifers ($b > 1$) where $Q_o$ increases with increasing initial saturated thickness (see Figs. 2 and 3 in Rupp et al., 2006b). Though the particular timescale is not important to our objectives, we chose it to be 45 days. Brutsaert (2008) noted a tendency for $\tau$ to be near 45 days across a large number of basins when fitting Eq. (1) with $b$ =1 to point cloud data. It remains to be seen whether a similarly narrow distribution of $\tau$ occurs for $b$ not equal to 1.

A pulse recharge amount corresponding to a given $Q_o$ can be calculated by integrating Eq. (2) from $t = 0$ to $t = \infty$. For $w > 1$ ($b < 2$), the recharge volume is

$$V = DA = \tau Q_o / (w - 1), \qquad\qquad\qquad (3)$$

where $D$ is the depth of recharge and $A$ is the aquifer area. For $w <= 1$ ($b >= 2$), integrating Eq. (2) results in an infinite volume, so $b > 2$ can only be sustained over a finite part of any recession. Values of $b > 2$ have been derived from physical theory for the early portion of a recession (Brutsaert and Nieber, 1977; Rupp and Selker, 2005) or can be obtained from recession curves over a finite time period while retaining physical realism by combining discharge from multiple linear ($b = 1$) or non-linear ($1 < b < 2$) reservoirs (e.g., McMillan et al., 2011). The effect on $b$ of combining linear reservoirs in parallel (e.g., Clark et al., 2009; Gao et al., 2017; Harman et al., 2009) and series (e.g., Rupp et al., 2009; Wang, 2011) has received much more attention.

We compared three hypothetical time series generated with different assumptions about the distribution of the magnitudes and inter-arrival times of recharge events and the superposition of recession events (Table 1). The inter-arrival times are distributed log-normally (Cases 1 and 3) or uniformly (Case 2). Event magnitudes (as defined given by $Q_o$) are either distributed log-normally (Cases 1 and 3) or have constant magnitude (Case 2). Events are either independent of antecedent conditions (Case 1) or events are superimposed on antecedent conditions (Cases 2 and 3) (Table 1 and Fig. 2).

To generate the time series for Case 1 and 3, independent recessions were created using a random number generator for a lognormal distribution for event peak magnitude and duration for a total of 10 years of time-series data. The lognormal distributions for event magnitude and duration are chosen for the synthetic hydrographs because the distributions for Lookout Creek are skewed right and roughly lognormal which is also consistent with other skewed right precipitation distributions in previous studies (Begueria et al., 2009; Selker and Haith, 1990). Recharge events were created with log-normally distributed inter-arrival times ($\mu = 2.5$, $\sigma = 1$) and event magnitudes ($\mu = 1$ day, $\sigma = 1$) where both values are normalized by timescale and the unit hydrograph respectively, resulting in dimensionless quantities. These values of $\mu$ and $\sigma$ result in event lengths with a mean of 20 days and an average of 18 events per year. This value was chosen to be comparable to the 19 events per year identified in the Lookout Creek example. The distributions of both the inter-arrival times and event magnitudes are skewed right, representing the high frequency of smaller events and less frequent large events. Changing $\mu$ and $\sigma$ will modify the amount of variability in individual recessions, and could be further explored with different distributions in future research regarding the resulting variability in $b$. Case 2 assumes constant event inter-arrival time ($\mu = 450/\tau$) and magnitudes ($\mu = 1$). The mean inter-arrival time of 10 days is intended to be comparable with the 19 events per year identified in the Lookout Creek example.

For Case 1, the individual recessions were combined to make a time series such that each event was concatenated onto the last event disregarding the antecedent flows. For Case 2 and 3, individual recessions were superimposed on antecedent flows,

appealing to the simplest model presented by the instantaneous unit hydrograph method (Dooge, 1973). We acknowledge that framework for the instantaneous unit hydrograph as described in Dooge (1973) does not consider non-linear reservoirs, but use it as a simple representation to produce variability between recessions; we discuss the implicit assumptions of this model in the Discussions and Conclusions section. From Fig. 2, the baseflow from the first event, $Q_B$, is a simple continuation of the first recession. The underlying second event, $Q_C$, is defined by the second event's initial magnitude (constant in Case 2 and randomly generated in Case 3). The resulting flow, $Q_D$, is the sum of $Q_B$ and $Q_C$.

As a result, Case 1 looks specifically at a time series events where the falling limb of each event maintains the same decay constant and the effect of having no antecedent baseflow influence on streamflow. By including baseflow to Case 2 but maintaining equal inter-arrival times and event magnitudes, we look specifically at the effect of antecedent conditions on individual recessions and the point cloud. Case 3 combines the distribution of event inter-arrival times and magnitudes of Case 1 with the baseflow conditions of Case 2, best representing the variability and inter-arrival times of individual recession events seen in Fig. 1 for data from Lookout Creek. Each case will address how the controls on the hydrograph affect the recession analysis plot and the estimates of *a* and *b*.

[Insert Fig. 2]

[Insert Table 1]

## 2.2 Recession Extraction Method

Recession extraction from observed hydrographs and the associated sensitivities to different criteria has been explored by Dralle et al. (2017), including minimum recession length, and the definition of the beginning and the end of the event. For Lookout Creek, we used extraction criteria similar to those of other studies (e.g., Chen and Krajewski, 2016; Dralle et al., 2017; Stoelzle et al., 2013) and applied the same criteria prior to all fitting methods presented in Section 2.3 to isolate differences in calculated *b* values due to fitting method only. An individual recession event duration must be longer than 5 days. Rainfall data can be used to identify non-interrupted recessions, but rainfall data we will not available in all cases, so we rely on the hydrograph only. The start of the recession is defined as one day after the discharge peak to account for the presence of overland flow. The end of the recession occurs at the minimum discharge prior to an increase in discharge greater than the error associated with instrument precision for stage height of ~0.01 ft, which translates into errors in discharge from ~0.01-0.1 $m^3$/s depending on the rating curve and the discharge level (Thomas et al., 2015).

For the synthetic hydrographs used in Section 3.2, events of any length were included, the recession start was selected at peak discharge because overland flow was not a factor, and the end of the recession was chosen as the time immediately before the next generated discharge peak.

## 2.3 Parameter Estimation Methods

Four methods of estimating representative recession parameters were evaluated: lower envelope (LE), central tendency (CT), binning average (BA; Kirchner, 2009), and the median of individual recessions (MI) (Roques et al., 2017). Linear regression in bi-logarithmic space was used with each method for consistency across methods.

Because a change of hydraulic regime was suggestive in Fig. 1 between high flow ranges and low flow ranges, recession analysis parameters were estimated for two flow ranges, early-time and late-time. Early-time and late-time describe a theoretical transition of flow regimes between high-flow and low-flow ranges (Brutsaert and Nieber, 1977). To reduce the subjectivity of distinguishing between high and low flows, a breakpoint in discharge separating high from low flow behavior was optimized to best represent the analytical solution. By separating the data into two subgroups, either smaller or larger than

a defined breakpoint discharge, the best fit line was determined for each subgroup. The location of the breakpoint is defined so the error between the observed ratio of $b$ for the two subgroups and the theoretical ratio ($b$=3 for early and 1.5 for late give a ratio of 2) is minimized, theoretically defining the subgroup above the breakpoint as early-time and the subgroup below the breakpoint at late-time.

For each of the four estimated methods, parameters were estimated for the early-time and late-time behavior separately. For

the LE method, $b$ was fixed to 3 and 1.5 for early and late-time, respectively, following Brutsaert and Nieber (1977) and $a$ was chosen such that 5% of points were below the lower envelope (Brutsaert, 2008; Troch et al., 1993; Wang, 2011). It should be noted that using these values for $b$ assumes that the groundwater discharge behaves like discharge from a single, initially-saturated, and homogenous Boussinesq aquifer. An alternative method to fitting the lower envelope without a pre-defined value of $b$ was introduced by Thomas et al. (2015) using quantile regression to estimate both $a$ and $b$ but was not used in this

study. For the CT method, the fit included all -$dQ/dt$ vs $Q$ points unweighted (Vogel and Kroll, 1992). For the BA method, bins spanned at least 1% of the logarithmic range, and a line, instead of the polynomial suggested by Kirchner (2009), was fit to the binned data. We dispensed with the inverse-variance bin weighting used by Kirchner (2009) to account for data noise when we applied the method to the synthetic recessions because some bins contained few points with very low variance and therefore were weighted excessively. For the MI method, parameters were estimated for individual recessions following

Roques et al. (2017) and the medians of $a$ and $b$ were calculated. In all cases, the time derivative -$dQ/dt$ was computed using the Exponential Time Step method (ETS) proposed by Roques et al. (2017).

## 3 Results

### 3.1 Parameter Estimation for Observed Recessions (Lookout Creek)

In Fig. 3 we display the recession plot stacking all individual recession resulting in the formation of the point cloud. The different fitting strategies revealed that the LE, CT, and BA methods all fit to the point cloud and result in different values $b$ when applied to the observed daily averaged streamflow for Lookout Creek: Early-time values of $b$ are over 50% larger for LE (fixed at 1.5) than CT and BA, and late-time values of $b$ are 50% and 25% larger for LE than CT and BA, respectively (Fig. 3 and Table 2). The CT and BA methods are fairly consistent with each other for both early and late-time, whereas the pre-defined theoretical $b$ values for the LE appear to provide poorer fits to the point cloud.

More importantly, parameter estimation differs greatly whether the point cloud or individual recessions are used. The late-time $b$ value which defines the low-flow baseflow regime is 6 times greater for MI than CT (Table 2). Using the MI method, the $b$ value is larger than any other method for both early and late time.

[Insert Fig. 3]

[Insert Table 2]

**3.2 Synthetic Hydrograph Results**

Based on the similar results from BA and CT methods discussed above, and the questionable practice of setting an early- and late-time $b$ a priori as we did in the LE method, hereafter we use the BA method to represent to point cloud recession parameter estimation when comparing to the MI method using individual recessions.

The recession decay exponent $w$ in Eq. (2) was set to 1.2; distinct values of $w$ were not used for early and late time. This value for $w$ results in $b = 1.8$ for an individual synthetic recession, which is near the reported median of individual $b$ values of 2.0 in Biswal and Marani (2010), and 2.1 in Shaw and Riha (2012) and Roques et al. (2017), though less than the median individual $b$ of 2.8 for Lookout Creek.

The $b$ values and the offset of individual recessions resulting from Eq. (1) are both functions of $w$. A larger $b$ value indicates a more stable baseflow discharge (a slower decline rate for given discharge). For a given value of $b$ and $\tau$, $a$ varies inversely with $Q_o^{1/(b-1)}$. Decreasing $w$ results in larger values of $b$ while also increasing the offset between individual recessions, resulting in a larger range of $a$ values and a more scattered point cloud. In contrast, as $w$ approaches infinity, the offset is minimized as $b$ goes to 1, representing in an exponential falling limb recession in time (Rupp and Woods, 2008). In this special case, all of the individual recessions overlap with constant $a$ (i.e., there is no offset among individual recessions lines). While $b=1$ is interpreted as a linear reservoir according to traditional theory and is a convenience often assumed, this result suggests that a condition where $b=1$ and $a$ is a constant is not consistent with the existence of a point cloud, except to the degree at which observation error introduces noise into the recession hydrograph, or other pathways (e.g., overland flow) contribute to the flow in the stream. In summary, the more linear the response is (the closer $b$ is to 1), the smaller the offset, whereas the more non-

linear the response (the larger the $b$), the greater the offset will be and thus the more different the parameter estimations will be between the point cloud and individual recession methods. The 3 following cases using synthetic hydrographs are intended to highlight the offset of the individual recession curves.

### 3.2.1 Case 1

Recession analysis of a hydrograph with log-normally distributed event inter-arrival times and peak discharge with a constant falling limb decay constant (no baseflow represented) results in individual recession events with the same $b$, horizontally shifted based on the initial discharge (Fig. 4). In this case, the peak flow of the event is the only source of variability in the recession parameter $a$. The variable event magnitudes result in individual events located over a range of $\ln(Q)$ values whereas if the same flow magnitude was preserved for each event, each individual recession would plot on top of one another creating

a single line without a point cloud. The variable event inter-arrival times change the duration of an event, with longer events occurring over a greater range on the y-axis. In this simple hydrograph, parallel individual recessions are present all with $b =$ 1.8, as expected. The value of $b$ is estimated at 1.3 considering the point cloud, which appears to be significantly less than imposed individual $b$ value of 1.8. This underestimation results from the offset between individual recessions based on the range of initial discharges.

To examine the sensitivity of parameter estimation to recession extraction criteria, we evaluated how choosing the start of the recession (i.e., the time elapsed since peak discharge) affects the value of $a$ when using the point cloud method. Whether we chose 0, 1, or 2 days following peak discharge, $a$ from the point cloud $a$ was 0.17 [-] and $b$ was 1.3 [-].

[Insert Fig. 4]

### 3.2.1 Case 2

The superposition of recession events accounts for the effects of antecedent baseflow. The superposition changes the effective $w$ of the falling limb of the hydrograph as the event recession is added to the antecedent events, resulting in variable $b$ across the individual recessions (Fig. 5). The median $b$ represented is 3.25 with a range of 2.56 to 3.41 (quantile range represented in the colorbar of Fig. 5). The point cloud $b$ of 2.35 falls outside of the range of $b$ values for individual recessions. Superposition results in a larger $b$ than what would arise from non-superposition. Steeper recessions (higher $b$) are associated with events

with higher baseflow contribution given the same addition of flow. By including antecedent flow conditions, neither $a$ nor $b$ is preserved between individual recessions.

[Insert Fig. 5]

### 3.2.1 Case 3

A hydrograph more representative of real-case conditions includes variable inter-arrival times and event magnitudes from Case 1 and baseflow antecedent conditions from Case 2 (Fig. 6a). These complexities result in a recession plot where the individual recessions represent the variability in watershed response represented by the hydrograph (Fig. 6b), where $a$ and $b$ are different between individual recessions. As with Case 1 and 2, the median individual $b$ (3.3) is greater than the point cloud $b$ (2.0) The

minimum individual $b$ is 1.9 with a maximum of 8.5 while the point cloud $b$ is near the low end of the range of individual $b$ values (See colorbar of Fig. 6). The similarity of features of Fig. 6b and Fig. 1 are noteworthy. Though many of the observed recessions in Fig. 1 are slightly curvilinear (in the log-log space) whereas the synthetic recessions are power laws, in both cases there is a tendency for recessions with lower initial discharges to have higher values of $b$ yet still many instances of recessions with similar initial discharges but different values of $b$.

[Insert Fig. 6]

## 4 Discussion and Conclusions

In the 42 years since Brutsaert and Nieber (1977) proposed their recession analysis, it has provided a seemingly simple analytical method for estimating basin-scale hydrologic properties. However, recent studies have highlighted the sensitivity to estimation methods on the recession parameter values and to the resulting interpretation of average watershed behavior. This

paper explores the effect of the distribution (in time and in magnitude) of individual recessions on parameter estimation and compares that to the parameter estimation from collective recessions (i.e., the point cloud). The four estimation methods considered were the lower envelope, central tendency, binning, and individual recession method. Because of the poorer apparent fit and problems pointed out from previous studies when using the lower envelope and central tendency methods, we chose to use the binning method to compare with results from the individual recessions method for a set of synthetic case

studies.

We hypothesize that the climate controls the distribution of individual recessions in bilogarithmic plots of $-dQ/dt$ vs. $Q$. This distribution can be related to the variability in recession analysis parameters. Using the three synthetic case studies, we examine the effects of event inter-arrival time, magnitude, and antecedent conditions on the distribution of individual recession events that comprise the pattern of collective recessions.

We conclude that recession analysis performed on collective recessions does not capture average watershed behavior, regardless of the fitting method. The point cloud is an artifact of the variability of the individual recessions, including the event inter-arrival times and distribution of magnitudes. Individual recessions with the same $b$ but different $a$ can be produced by varying the initial discharges (Case 1), variability of $b$ for individual recessions can be produced by superimposing events on antecedent flow conditions (Case 2), and different recession event lengths with different $b$'s can be produced by including

variable event inter-arrival times and magnitudes (Case 3).

For Case 1, the recession analysis parameter $a$ is equal to $w/(\tau Q_o^{1/w})$ and thus the intercept of the individual recession curves will scale with $Q_o$. The result is a collection of individual recession curves that are horizontally offset based on the initial discharge producing a smaller $b$ value for the point cloud compared to the individual recessions. Case 1 illustrates that the slope of individual recession events can be greater than the best fit line through the point cloud, consistent with previous studies (Biswal and Marani, 2010; Mutzner et al., 2013; Shaw and Riha, 2012). However, the point cloud in Case 1 is generated by a collection of multiple individual recessions all with the same slope and does not have the variability in $b$ values presented in these previous studies and shown for Lookout Creek in Fig. 1. Case 2 and 3 use superposition of antecedent flow events that consequentially changes the individual $b$ values, providing a possible explanation for the variability in $b$ values for individual recessions. For Case 2, the superposition of events takes account of antecedent conditions which results in a distribution of individual $b$ values despite the decay exponent $w$ being fixed. For Case 3, the horizontal offset of individual recessions from Case 1 and the effects of antecedent conditions from Case 2 result in the recessions with variable of individual $b$ values and that are horizontally offset to create a pattern similar to that observed in a real watershed.

While the mean $b$ for individual recessions in Case 1 is a direct consequence of the value of $w$ used in Eq. (2), this is not true when the discharge from each application of Eq. (2), which we call an 'event', is superimposed on the antecedent flow, as in Case 2 & 3. This superposition of events results in a range of individual recession $b$ as often observed in the literature (Basso et al., 2015; Biswal and Marani, 2010; Mcmillan et al., 2014; Mutzner et al., 2013; Shaw and Riha, 2012), thus it appears that the straightforward superposition of events can recreate the watershed behavior. However, there is a key underlying assumption of this superposition that is inconsistent with a real watershed. To help describe this inconsistency, we can compare two distinctly different conceptual models of watershed. The first, and very frequently used, model is a single bucket with an outlet near the bottom. The bucket contains a porous medium whose properties may vary with depth to create a variety of non-linear outflows. Each new recharge event adds to the pre-existing storage of water in the bucket. The second model is the one used for Case 2 & 3: each new event adds water to a new and independent bucket and the outflows from all buckets are aggregated. Both conceptual models have components that are patently unrealistic when applied to natural watersheds but, remarkably, the latter model produces a distribution of recession events in $-dQ/dt$ vs. $Q$ space that is more like what is observed in Lookout Creek basin and others (Mutzner et al., 2013; Shaw and Riha, 2012). This finding reveals key information about the subsurface plumbing system of the basin and its dynamics that could be explored with models with a higher degree of realism. We acknowledge that there are other ways to create watershed memory that would also generate variability in apparent recession parameters that would be worthwhile to consider. For example, following previous works that have shown that multiple linear reservoirs can generate power-law recessions (Clark et al., 2009; Harman et al. 2009), one could explore combinations of parallel linear reservoirs with varying sizes and recession constants under time-varying recharge. However, based on the results of Harman et al. (2009) using periodic recharge events, it is not clear that this would lead to a distribution of recession curves of varying b values like what is seen in Figure. 1. A similar, albeit more complicated, exercise could also be done with combinations of parallel non-linear reservoirs with distinct recessions parameters.

An additional important simplifying assumption of this study is the use of a constant timescale $\tau$ for each individual event. Previous studies that have examined timescales across basins by setting $b = 1$ and estimating $\tau$ from the point cloud (Brutsaert, 2008; Lyon et al., 2015). However, given the questionable of the validity of the point-cloud estimation methods, additional studies of the variability of timescale among individual recession events and across basin should be done.

We show how the point cloud pattern does not arise from watershed properties alone. The consequence is that parameters estimated from the point cloud do not represent watershed properties. In all three synthetic hydrograph representations, the median individual recession $b$ is significantly greater than $b$ from the point cloud. Additionally, it is possible for the point cloud $b$ to be smaller than the minimum individual recession $b$ indicating the point cloud fit represents a behavior outside the range of watershed responses represented by individual recession events. In contrast to the point cloud, individual recession

analysis provides insights into the average and variability of watershed responses which is highly dependent on the memory effect of the watershed. The variability in individual responses gives insight into watershed complexities including heterogeneity in topography, geology, and climate. Watersheds may present large variability in geology and so hydrogeological conditions such as unconfined/confined aquifers, inter-basin groundwater flows, high spatial hydraulic conductivity variability, depth-dependent hydraulic conductivity, or large-scale discontinuities. As a result, there are still opportunities to further

characterize the variability in watershed responses and the associated variability in individual $b$ values to improve streamflow prediction using recession analysis.

A strength of the critical zone community is the ability to create a global analysis by comparing across studies (Brooks et al., 2015; Fan et al., 2019). However, a lack of consensus for a standard method for recession analysis procedures exists and thus inhibits recession analysis studies from being widely compared. If streamflow analysis is to be included in a global analysis,

results need to be comparable across scales and observatories. There is a need for a common method employed to compare the average and variability in watershed responses. Because estimated parameters may differ greatly by estimation method, misinterpretation of hydrological properties and incorrect predictions within the critical zone are possible. When using the point cloud in particular, a smaller recession parameter $b$ at late time could be, and has been, interpreted to imply greater basin vulnerability to drought (e.g., Berghuijs et al., 2016; 2014; Yeh and Huang, 2019). However, a more stable baseflow is implied

by distribution of $b$ from the individual recessions and its median $b$ which can be much larger than what is estimated from the point cloud. We suggest that the use of collective recession analysis should be avoided in favor of individual recession analysis as the standard to describe the average and variability in watershed response. The methods employed for recession analysis certainly require more attention: Correct methods are critical to understand the underlying hydrology and thus the interpretation of a watershed's vulnerability to climate change.

**Code and Data Availability**

Streamflow record for Lookout Creek is freely available from the USGS website. Source code for the exponential time-step method is available by request (Roques et al., 2017). Randomly generated log-normal event magnitudes and inter-arrival times presented     in     this     paper     for     Cases     1     &     3     are     available     at:

http://www.hydroshare.org/resource/e3c159631acd470cbeef5fa1abe0142e. Respective codes can be obtained from the corresponding author.

**Author Contribution**

Elizabeth R. Jachens, David. E. Rupp, and John S. Selker were involved in conceptualization. Elizabeth R. Jachens and Clément Roques developed the methodology and performed the analysis. Elizabeth R. Jachens prepared the manuscript with

contributions from all co-authors.

**Competing Interests**

The authors declare that they have no conflict of interest.

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

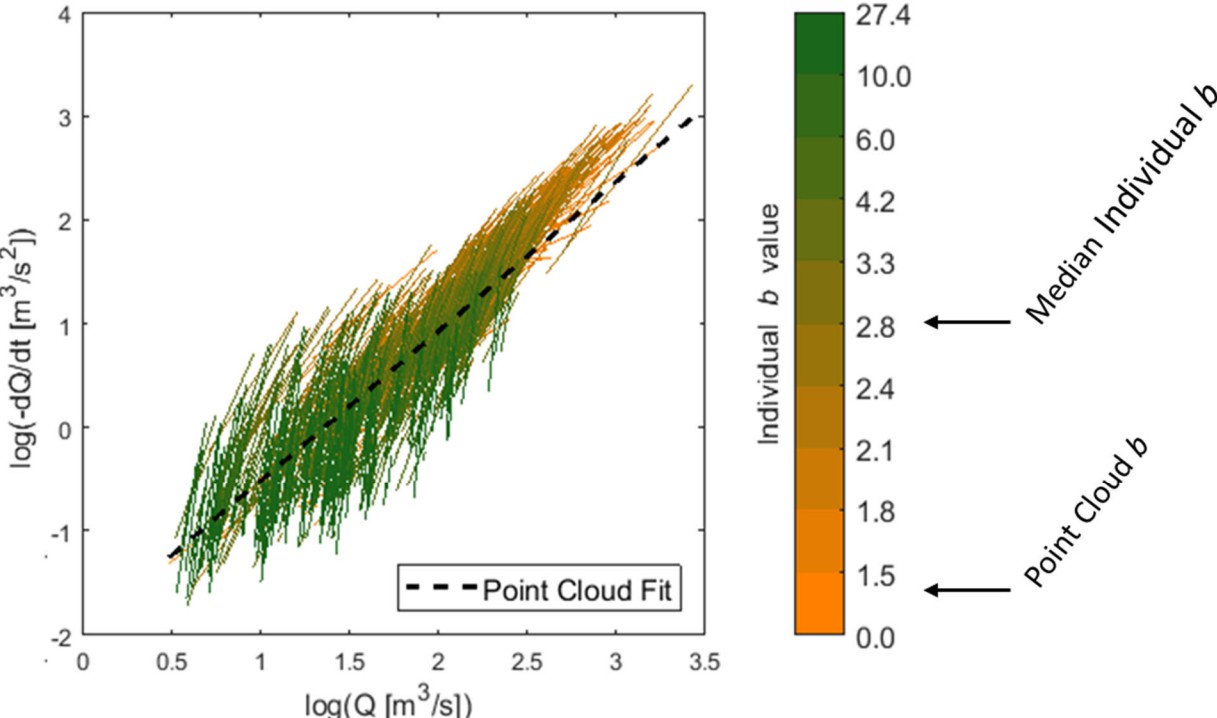

**Figure 1.** Recession analysis plot in log-log space for Lookout Creek (USGS# 14161500). Individual recession fits are displayed with color scale differencing by values following a discretization according to decile groups. This discretization allows the description of the organization of individual recessions where recessions with similar $b$'s that appear to be horizontally offset. The point cloud has $b = 1.4$ (binning average shown as a black dotted line) compared to $b = 2.8$ for the median individual recession.

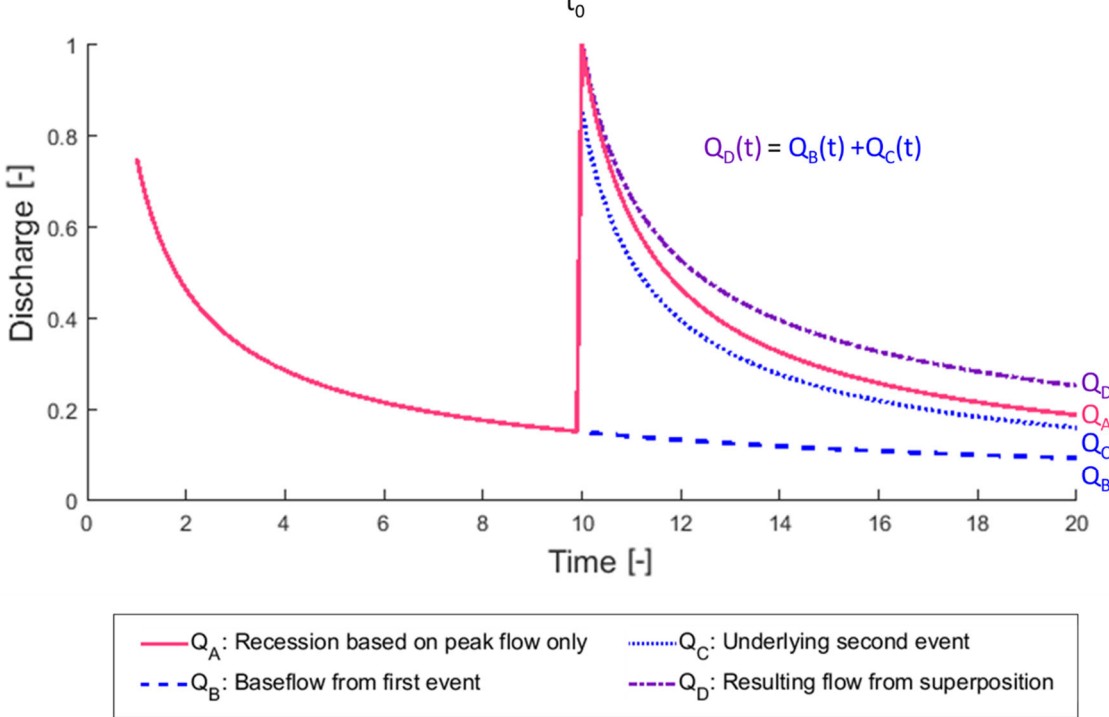

**Figure 2.** Conceptual model of identical recession events only dependent on the initial dimensionless flow ($Q_A$ representing Case 1) and superposition of events to include antecedent conditions (summation of the blue dotted ($Q_C$) and dashed line ($Q_B$) resulting in the superposition of the flow in the purple dash-dot line ($Q_D$ representing Case 2&3)). By superimposing the antecedent flows ($Q_B$) on the underlying event ($Q_C$), the effective falling limb ($Q_D$) is less steep than the non-superimposed falling limb ($Q_A$). Time is expressed in dimensionless units and arbitrary values.

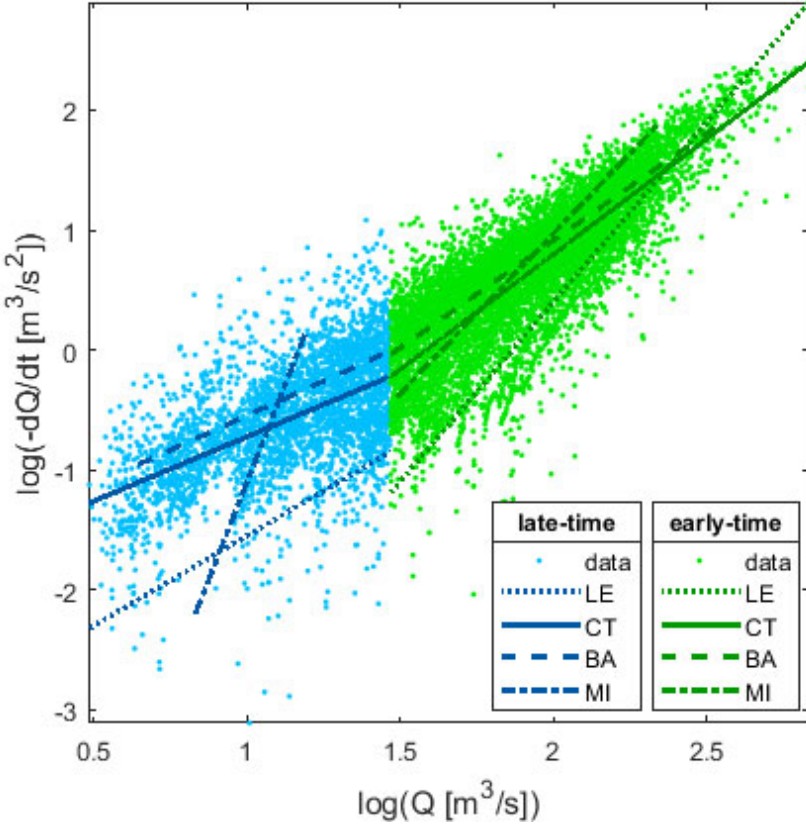

**Figure 3.** Recession analysis for Lookout Creek to aid in the comparison of four different fitting methods and the dependency on parameter estimation shown visually (lower envelope (LE), central tendency (CT), binning average (BA)) and individual recessions parameters (median individual recession (MI)). Depending on the fitting method, the parameter estimation for *a* and *b* will be different.

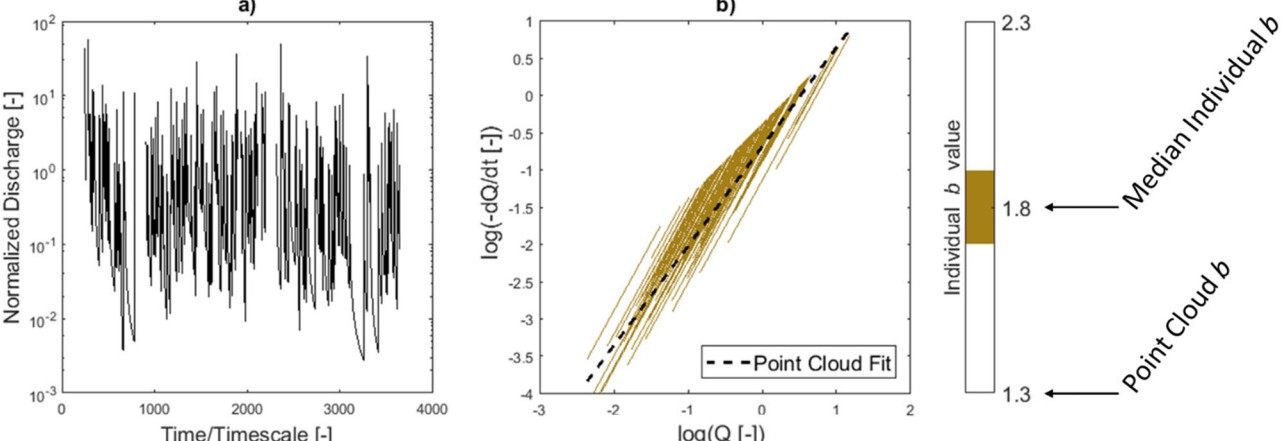

**Figure 4.** a) Hydrograph with log-normally distributed event inter-arrival times and peak magnitudes with each event maintaining a constant falling limb decay constant, and b) recession analysis with resulting parallel individual recessions having a constant *b* value (MI *b* = 1.8) compared to the point cloud fit (black dotted line) which results in *b* = 1.3). Discharge and time are normalized, resulting in dimensionless quantities. Gaps in the hydrograph are a result of individual event magnitudes that are smaller than the streamflow that precedes the event start. The individual recessions are offset which when viewed collectively, results in the point cloud.

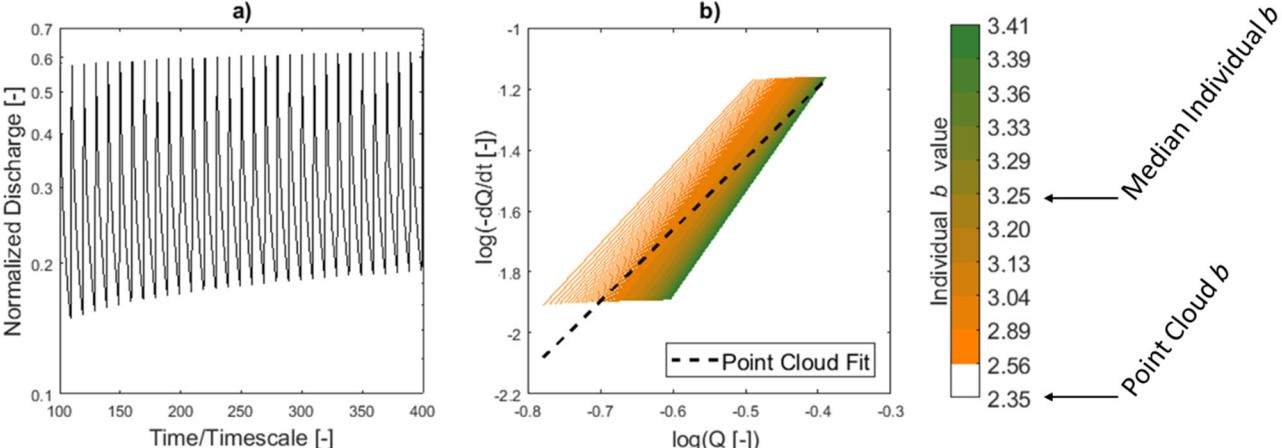

**Figure 5.** a) Hydrograph of equally spaced recharge events with each underlying equal magnitude recession event superimposed on previous ones resulting in varying falling limb decay constant (x-axis zoomed in to show detail), and b) recession analysis plot showing a range of *b*'s of individual recessions (MI *b* = 3.25), with steeper recessions associated with events with higher baseflow contribution, compared to the point cloud fit (black dotted line- BA *b* = 2.35). The color bar is divided into 10 ranges based on the individual *b* value, each range contains 10% of individual recessions, and the lowest range in white for comparison to the point cloud range.

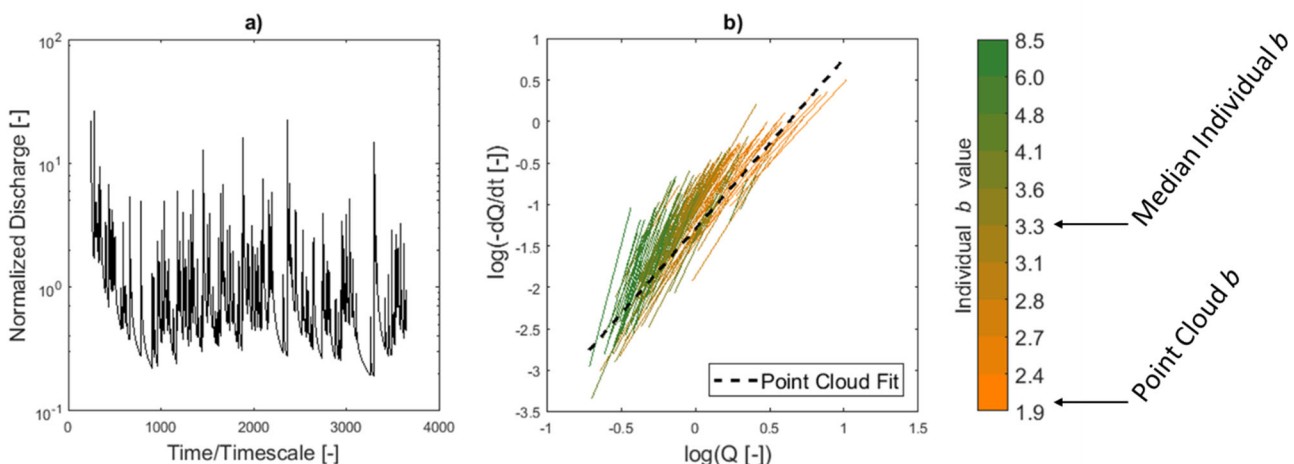

**Figure 6.** a) Hydrograph with lognormal distribution in recharge event inter-arrival times and magnitudes, and b) recession analysis plot showing a large range of *b* values (with the median of *b* = 3.3), compared to the point cloud fit (black dotted line- BA *b* = 2.0). The color bar is divided into deciles in the distribution of b values compared to the point cloud range.

**Table 1.** Synthetic hydrograph scenarios

|        | **Event Magnitudes** | **Event Inter-Arrival Time** | **Superposition of antecedent flow?** |
|--------|-----------------------|------------------------------|---------------------------------------|
| **Case 1** | Log-normal | Log-normal | No |
| **Case 2** | Constant | Constant | Yes |
| **Case 3** | Log-normal | Log-normal | Yes |

**Table 2.** Comparison of recession analysis parameters *a* and *b* for Lookout Creek between different methods: lower envelope (LE), central tendency (CT), binning average (BA), and the median individual recession (MI). Each value is represented as a

ratio of parameter estimation for early to late time.  Depending on the fitting method, the parameter estimation for *a* and *b* will be different

| | $log(a$ $[s^{-1}(m^3/s)^{1-b}])$ | | $b$ [-] | |
| --- | --- | --- | --- | --- |
| | early | late | early | late |
| LE | -5.6 | -3.0 | 3.0 | 1.5 |
| CT | -3.0 | -1.8 | 1.9 | 1.0 |
| BA | -2.6 | -1.6 | 1.8 | 1.2 |
| MI | -3.9 | -8.1 | 2.7 | 6.4 |

