# Peer review of "Recession analysis revisited: impacts of climate on parameter estimation"

_Hydrology and Earth System Sciences, 2019_

## Referee Comment (RC1) · Michael Stoelzle (Referee) · 28 Jun 2019

I like the paper (i.e. the research question, methods, analyses) and it is in the scope of HESS. The authors investigated a common technique in recession analysis (dQ/dt-Q- point clouds) and argued that parameter fitting based on this cloud is misleading to better understand catchment functioning. Instead, in line with other recent publications, the authors suggest that individual recessions (i.e single events) should be analyzed. The paper is rather short what is good and written in a good style. I suggest minor revision before publication.

**Major comments**

1. I have some concerns about the title of the paper. It is not really specific and the "42

years" will be only valid for this year 2019, that is not a good choice, I think. It would be better to include the difference between analyzing dQ/dt-Q-point clouds and individual recession. By the way, the paper is not about "recession analysis" (as there are many other recession analysis techniques out there, e.g. master recession curve), so the focus on the dQ/dt-Q-cloud could be emphasized in the title.

2. Is the difference between slope b from the point cloud and slope b from median individual b a function of the recession extraction criteria (e.g. longer than 5 days, cut the first day, etc., P02L26). By the way, have the recession a minimum of 5 days with or without the removed first day? Brutsaert and Nieber 1977 also used rainfall data to extract recession segments... The recession extraction should be clearly stated in the methods (not only in the Introduction).

3. median parameters from individual recession analysis (P03L04): I missed a discussion about seasonal catchment or streamflow behavior in the paper. There are many studies with recession analysis considering seasonal components in a, b, and dQ/dt-Q-plots. The authors should at least refer to those studies or give a comment on the issue.

4. Why synthetic hydrographs (P03L08)? Advantages of this approach should be (shortly) mentioned (earlier than in Sect. 2.2).

5. Section 3 is Results, not Methods

6. "The pre-defined theoretical b values for the LE appear to provide poor fits for the point cloud" (P05L17) very important results for all further studies that using LE and pre-defined b's. You should emphasize this (even more)!

7. more explanation on the hydrological functioning of a and b (or give references), like e.g. P07L11, this would make it easier for a broader readership.

8. explain in more detail how the synthetic hydrographs are derived (or give references). Why is this way better than using multiple (real world) catchments?

9. How representative is the median b from individual recessions across all Cases 1-3 for the individual recession hydrographs? It might be beneficial to show the distributions of b around the median (e.g. violin plot?)

10. I asked myself what is the most sensitive part of the recession analysis presented here? The recession extraction method? The fitting procedure? Are there specific references discussing the different sensitivities? That means the value of the paper could further be improved if the authors give some guidance on how specifically the individual recession segments should be extracted then.

**Minor comments**

P01L21 Here the Brutsaert and Nieber and the Kirchner paper are missing.

P01L24 Add Brutsaert and Nieber, 1977 as a first fundamental paper using dQ/dt-Q.

P02L03-07 Better change the two sentences: first collective recession analysis remains common, then the recent criticism.

P02L10 Lower envelope has not been introduced yet.

P02L20/21 Explain shorty what a greater b means in terms of recession behavior.

P03L11 Last sentence of section 1 is rather a result or a conclusion, but can be rewritten to a hypothesis or research question.

P03L14 I would change the order; first the definition of hydrographs then the parameter estimation methods (which needs the hydrographs to be applied).

P03L24 Would be helpful to give key references here for the four fitting methods and to state if the original procedures from the references are used or modified.

P03L27 Why 5% (and not 10%), is 5% common?

P06L18 Last paragraph of section 3.2 belongs rather to the methods than to the results section. At least, a reference in the methods section referring to this result should be

given.

P08L21:"sum of the squares", I don't get this.

**Technical comments**

P01L13 "based on dQ/dt-Q-recession analysis"

P02L01 why "primary"?

P02L14: two times Kirchner reference, remove one.

P02L24 "significantly"? sound like it was statistically proven?

P03L05 "Being able to... vulnerability studies" - the sentence is not well connected here, move up?

P03L22 "theatrical"?

P08L22 to understand the underlying hydrology

P08L23 Sentence is a repetition of the last sentence of the paragraph before

---

## Referee Comment (RC2) · Anonymous Referee #2 · 12 Jul 2019

——Paper summary——

The authors explore sources of variability in the fitted parameters of the power law streamflow recession model: $dQ/dt = -aQ^b$ by generating collections of synthetic recessions. Three cases are examined to investigate the various ways one might obtain conflicting results when comparing individual event vs. point cloud methods.

——General Comments——

While I agree with the authors that there is "work yet to be done" in the field of streamflow recession analysis, and really appreciate some of the authors' discussion points on the practical implications of using point cloud analysis vs. individual event analysis (Page 8), there are a number of parts in this manuscript that I find difficult to understand, or that I believe impose "baked in" sources of variability that may not reflect the forms of variability imposed by actual physical processes.

1) In some respects, it seems that Case 1 really encapsulates the main point of this paper (which numerous authors have already argued; though I think the point is worthy of reiteration); that individual recession events typically have steeper slopes than the best fit slope of a line through a point cloud generated by a collection of individual events. Case 1 demonstrates how this might happen; if the recession scale parameter (a) scales with the initial flow condition of the recession (Q0), the intercept of the recession curve in log-log space will shift up or down with Q0, and so a collection of steep recession curves will "stack" in a such a way as to create a point cloud that is less steep than the individual curves of which it is composed (see specific comment #5 for additional comment on imposing this form of variability in 'a'). Cases 2 and 3 are primarily used in the Discussion and Conclusion to demonstrate this same point. For this reason, I do not see how these cases are useful. These cases might be advantageous if the authors were able to systematically explore the effects of the magnitude-frequency distribution of recharge events on individual recession curves (for example, convincingly attributing the spread in "b" to the magnitude-frequency statistics of recharge). However, I would argue, there is no clear way to perform such a systematic exploration given our present understanding of the physical origins of power law recession dynamics.

2) The methods need to (a) more thoroughly explain exactly how to generate the various forms of synthetic recession, and (b) how these different forms might reflect the impacts of real, physical processes in a watershed. On the first point:

(a)

i. How do the authors translate a recharge magnitude (presumably with units of [L]?) into a flow increment (with units of [L/T])? In the case of nonlinear recessions, the flow increment is a nonlinear, flow dependent function of the recharge depth.

ii. What are the parameterizations used for the various distributions from which flow increments and inter-arrival times are sampled?

iii. How am I to interpret the .mat file uploaded to Hydroshare? I loaded this file, and I see there are columns "mag", "start_locs", and "value". How do I use this information to reconstruct the recession curves the authors analyzed? It's undocumented, and not described in the text.

iv. For Cases 2 and 3, can the authors more clearly define their superposition procedure? Going off of Figure 2, how is the "underlying second event" (QC) constructed? Is the recharge increment added to the value of flow at the end of the previous recession? Or is this how QA is generated? One possibility for QC (once the authors clarify how it is constructed) is that we have effectively created a second "reservoir" with an initial storage equal to the magnitude of the recharge event. Then, QD would equal the sum of the discharge from the continued first event and the discharge from the second reservoir.

(b)

Referring to comment (iv.) above, it's not clear how this appearance of a second reservoir represents any physical process, or why it's a meaningful way to generate variability. The idea that the previous event recession somehow continues unabated and superposed with the current event effectively splits the watershed into two parallel components that, owing simply to the occurrence of a recharge event, now operate independently of one another. The procedure amounts to taking the sum of two nonlinear reservoirs with identical values of 'b' (page 5, Line 23), and varying value of 'a' imposed by Page 4, Line 17. I don't disagree that this will generate a new recession curve with entirely different power law parameters which depend on previous flow conditions, but the authors do not provide a rationale for imposing this form of memory. A more defensible approach (in my opinion) taken by previous authors is to explicitly acknowledge physical mechanisms that might give rise to parallel reservoirs throughout a landscape

(for example, conceptualizing a watershed as a collection of contributing hillslopes with varying hydraulic response times). In such cases, parallel reservoirs may generate increased nonlinearity, as demonstrated by Harman et al (2009) and Gao et al (2017). While it is true that these previous authors use superposition of linear reservoirs, the actual dynamics that give rise to increased nonlinearity are similar to those operating in the present work. On a related note, I think the authors should be citing these previous manuscripts, which I believe are very closely related to the present work.

3) I do not understand the purpose of the "early" vs. "late" fitting method in the context of this work. The early/late time methodology derives from the analysis of Brutsaert and Nieber [1977], who show that a shift from a recession slope of 3 to 1.5 is a direct consequence of the dynamics of a Boussinesq-style hillslope groundwater table. The physical implications of the authors choices in construction of synthetic hydrographs (e.g. existence of parallel reservoirs in the previous comment) are not necessarily consistent with the dynamics of a single hillslope groundwater table, so why use a form of analysis that is specific to the Boussinesq framework?

————Specific comments————————

1) Page 2, Lines 16 – 18: Do the authors intend to say that sources of variability in a,b between events may derive from these sources? Also, it is not clear what the authors mean by "flow superposition from previous events".

2) Page 2, Line 29: This statement is vague; of course the hydrology of a recession event affects the recession event.

3) Page 3, Line 9: Use of superposition without defining the term.

4) Page 3, Line 22: "theoretical"

5) Page 4, Line 15 – 17: While I agree that this is certainly one way to introduce variability in the recession scale parameter, it is nevertheless arbitrary to impose this particular relationship pinned to a 45 day timescale. Subsequent interpretation should be qualified with "where a = -w/(t0*Q0^(1/w)) holds...". While it is a convenient expression for imposing variability in 'a', I am unaware of any process-oriented result that shows the recession scale parameter should be determined in this way. Related to this, on Page 6, Lines 3 – 12, this discussion is difficult to follow. I think the authors are making the point that within their imposed timescale framework, the recession scale parameter 'a' must collapse to a single value that no longer depends on the flow initial condition in the limit as b=1. I agree that, mathematically, this is what happens, but the authors don't provide a compelling case that this is physically what should happen with real recessions; so the conclusion, "yet this result suggests that a condition where b=1..." should be qualified with the requirement that this would be true in circumstances where the authors' imposed form of variability for 'a' holds.

6) I assume the authors meant to put "3 Results" not "3 Methods" on page 5.

7) Page 7 Line 23: "We hypothesize..." is an almost tautological statement.

——-Figures————————-

- Figure 2: Why is there a "t0" at the top of the plot? Isn't t0 the 45 day timescale imposed to generate the recession scale parameter?

———————————————

---

## Author Response (AR1)

Dear Editor,

10

- 5 We appreciate the reviewers' comments that have provided a thorough review and constructive suggestions. In response to the reviewers' comments, we have implemented numerous changes to strengthen the manuscript. The major changes include:
  - 1. An updated title that does not include a time-dependent stamp
  - 2. Inclusion of an equation describing the relationship between discharge and recharge volume
  - 3. Updated figures to reflect the modified recession decay constant from 0.7 to 1.2 to better align with physical realism
  - 4. Additional details added to describe the synthetic hydrographs in the methods and the discussion, including a plot added to the supplemental materials
- 15 5. Reorganization of the methods section to increase clarity between methods applied to observed hydrographs and synthetic hydrographs
  - 6. Discussion on the physical representation generated variability in hydrograph responses

Please find below our responses to the comments from the two reviewers, as well as the

20 revised marked-up manuscript. We look forward to your evaluation and feedback on the revised manuscript.

Sincerely,

Elizabeth R. Jachens

**Major comments**

1. I have some concerns about the title of the paper. It is not really specific and the "42 years" will be only valid for this year

- 5 2019, that is not a good choice, I think. It would be better to include the difference between analyzing dQ/dt-Q-point clouds and individual recession. By the way, the paper is not about "recession analysis" (as there are many other recession analysis techniques out there, e.g. master recession curve), so the focus on the dQ/dt-Q-cloud could be emphasized in the title. Authors Reply: Thank you for this important point. We agree that the current title is limited by including the time stamp. We have greatly modified the title to reflect the motivation and conclusions presented in this paper. The updated title is: Recession
- 10 Analysis Revisited: impacts of climate on parameter estimation

2. Is the difference between slope b from the point cloud and slope b from median individual b a function of the recession extraction criteria (e.g. longer than 5 days, cut the first day, etc., P02L26). By the way, have the recession a minimum of 5 days with or without the removed first day? Brutsaert and Nieber 1977 also used rainfall data to extract recession segments. . . The recession extraction should be clearly stated in the methods (not only in the Introduction).

- 15 Authors Reply: In the revised paper we have included a new methods section (section 2.2) about parameter extraction methods where we have given more details. First, we have included citations of previous studies that have looked at the sensitivity of recession extraction on parameter estimation [*B. Chen & Krajewski*, 2016; *D. N. Dralle et al.*, 2017]. As this is not the primary focus of this paper, we have used criteria for event length and definition of the start and end of the recession similar to be consistent with previous studies. However, the sensitivity of extraction criteria on parameter estimation is valid consideration
- 20 and we have added to the paper. We have expressed a and b given the Equation of the falling limb of the hydrograph: "a=w/( $\tau Q_o^{1/w}$ )" and "b = (1+w)/w". Because b is only dependent on w, the individual recession b will be sensitive to the recession extraction for w. For a, recession extraction will affect  $Q_o$  and w. However, the offset will still occur which creates the difference between the individual recessions and the point cloud unless w goes to infinity. For Case 1, we changed the overland flow variable between 0, 1, and 2 days. This changed where the recession start was defined and thus  $Q_0$ . Given  $a=-w/(\tau Q_o^{1/w})$ ,
- 25 using overland flow of 0 days the range of a values for individual recessions is from 0.5 to 4.8 with a mean of 0.48. When overland flow is defined as 1 day (i.e. the first day of the recession is excluded in parameter estimation) the range of a values for individual recessions are from 0.5 to 5.4 with a mean of 0.50, When overland flow is defined as 2 days (i.e. the first two days of the recession is excluded in parameter estimation) the range of a values for individual recessions are from 0.5 to 5.6 with a mean of 0.50, For all three overland flow values evaluated, the point cloud a was 0.17. This discussion has been added
- 30 to the synthetic hydrograph results. Regarding the sensitivity of parameter estimation sensitivity, we evaluated how choosing the overland flow duration changed the value of *a* and found that the start of the recession to be relatively insensitive to *a*. Additionally, we have reworded the information about the minimum recession length to clearly define the minimum recession length before the overland flow cutoff is applied. Regarding the comment about rainfall, we have included a sentence

explaining that we did not use rainfall data for extraction criteria because a complete rainfall dataset is not always available for a watershed (while it is available for the watershed presented as an example in this study) and no rainfall record was created for the synthetic hydrograph. This allowed our methods for recession extraction to be consistent between the real-watershed example and the synthetic hydrographs.

5 3. median parameters from individual recession analysis (P03L04): I missed a discussion about seasonal catchment or streamflow behavior in the paper. There are many studies with recession analysis considering seasonal components in a, b, and dQ/dt Q-plots. The authors should at least refer to those studies or give a comment on the issue.

Authors Reply: We agree that a discussion on the literature looking at seasonality of a and b is important to this discussion. In the introduction, we have added a discussion of point cloud analysis that is sub-setting to months or season [*Szilagyi et al.*,

2007; *Thomas et al.*, 2015] and seasonal trends of a and b for individual recessions [*Bart & Hope*, 2014; *D. Dralle et al.*, 2015; *Karlsen et al.*, 2018; *McMillan et al.*, 2011; *Shaw & Riha*, 2012; *Tashie et al.*, 2019].

4. Why synthetic hydrographs (P03L08)? Advantages of this approach should be (shortly) mentioned (earlier than in Sect. 2.2).

Authors Reply: Synthetic hydrographs were chosen for this study because each individual recession can be definitively

- 15 identified as the characteristics are known which is unrealistic when considering real watersheds. Furthermore, the synthetic hydrographs can be specified to directly compare different hydrological controls without the confounding variables traditionally associated with real watersheds. For these purposes, the specifications of the synthetic hydrographs were chosen to explore the effects of the magnitudes and frequency of recharge events on the recession analysis parameters from collective vs. individual recessions. The justification for using synthetic hydrographs has been added to the methods section 2.1 under
- 20 the synthetic hydrograph methods.

5. Section 3 is Results, not Methods

Authors Reply: Corrected. Thank you.

6. "The pre-defined theoretical b values for the LE appear to provide poor fits for the point cloud" (P05L17) very important results for all further studies that using LE and pre-defined b's. You should emphasize this (even more)!

- 25 Authors Reply: We agree with this comment. While the poor fit of the LE is an important result, it is not new to our study. To give appropriate credit to previous studies that suggested the problems with the lower envelope method and to corroborate our claim, the sentence references have been expanded "... consistent with previous studies that have shown errors associated with the LE method (Rupp and Selker, 2006a)". We have added a sentence in the conclusions that discusses the 4 fitting methods considered and the apparent poor fit of the LE: "The four fitting methods considered were the lower envelope method, central
- 30 tendency, binning, and individual recessions. Because of the poor apparent fit and problems pointed out from previous studies, the lower envelope and central tendency were not considered in favor of improved methods for binning of collective recessions and the median individual recession."

Additionally, we have added a discussion about the limitations of using a pre-defined value for b to find the LE fit: "A limitation of using a pre-defined value for b for the lower envelope assumes that the watershed responds like a single homogenous

Boussinesq hillslope, which isn't known priori for the LE of a watershed composed of multiple heterogeneous hillslopes of unknown b."

7. more explanation on the hydrological functioning of a and b (or give references), like e.g. P07L11, this would make it easier for a broader readership.

- 5 Authors Reply: We appreciate this comment and agree that a practical interpretation of recession parameters was missing and should be included. To the introduction, we have included a description of the interpretation of b where a larger value of b indicates a decreasing streamflow decline and a smaller value of b indicated a faster rate of streamflow decline with decreasing streamflow. For a given value of *b*, a larger value of *a* implies a more conductive/permeable basin.
- For a large b value, at low discharges, the streamflow decline rate is less rapid than at larger discharges. As an extreme an 10 infinitely large b value will appear at a horizontal line on the recession analysis plot, indicating that the streamflow decline rate is zero and the stable streamflow magnitude would be the value where the individual recession crosses the x-axis. In contrast a b value of 0 would appear as a horizontal line on the recession analysis plot, indicating a constant rate of streamflow decline for all streamflows defined by the value at which the individual recession crosses the y-axis.
- In the results of the updated manuscript, we have included the explanation of different parameter estimations result in different 15 water management decisions. For the case studies when comparing parameter estimation for the point cloud vs individual recessions, if the point cloud b is smaller than the median individual b, using the point cloud would indicate a faster discharge decline rate than using the individual recessions and thus a more sensitive watershed to climate change.

8. explain in more detail how the synthetic hydrographs are derived (or give references). Why is this way better than using multiple (real world) catchments?

- 20 Authors Reply: Synthetic hydrographs are constructed using successive falling limb of the hydrograph following a power law. The three different cases are different in three ways: 1) the distribution of event magnitudes, 2) the distribution of event distribution, 3) the effect of antecedent conditions. Cases 1&3 have log-normally distributed inter-arrival times ( $\mu = 2.5$ ,  $\sigma = 1$ ) and event magnitudes ( $\mu = 1$ ,  $\sigma = 1$ ), compared to Case 2 with uniform event inter-arrival time ( $\mu = 450/\tau$ ), and magnitudes ( $\mu = 1$ ). The time series is based on concatenating successive individual events based on the guidelines for each case (see Table
- 1). We have also included a description of the superposition of events for Case 2 and 3 following the nomenclature from Figure2. We have included an expanded version of this explanation of how the synthetic hydrographs were created in methods section2.1. We have also included a more detailed description on Hydroshare where the time series for the synthetic curves is available.

We chose to use synthetic hydrographs instead of multiple real-world catchments because we were able to isolate specific
 climatic controls (event magnitudes, frequency, and superposition of events) between cases. As a result, we were able to specifically examine event magnitude, frequency, and the type of superposition of events in order to assess their influences on individual recessions. We can also determine the influence of parameter estimation for individual recessions and the point cloud. With real watersheds, there are many confounding variables that prevent the deconvolution of the climatic controls. We

have also commented on the advantage of using synthetic hydrographs compared to real watersheds in the response to Major Comment 4.

9. How representative is the median b from individual recessions across all Cases 1-3 for the individual recession hydrographs? It might be beneficial to show the distributions of b around the median (e.g. violin plot?)

- 5 Authors Reply: Thank you for the comment. We agree that the distributions of b around the median provide valuable information about the variability of watershed responses. We have chosen to represent the distribution of b in the colorbar for each RA plot using percentiles instead of using another graphic such as a violin plot. To increase clarify, we have added more definitive language describing the variability of b in reference to the colorbar of the Figures. For example, for case 2 "the median b represented is 3.25, with the range of individual b between 2.56 and 3.41 (quantiles represented in the colorbar of
- 10 Figure 5)"

10. I asked myself what is the most sensitive part of the recession analysis presented here? The recession extraction method? The fitting procedure? Are there specific references discussing the different sensitivities? That means the value of the paper could further be improved if the authors give some guidance on how specifically the individual recession segments should be extracted then.

- 15 Authors Reply: The reviewer brings up a good point. We acknowledge other authors have looked at the sensitivities of multiple parts of recession analysis including recession extraction [*B. Chen & Krajewski*, 2016; *D. N. Dralle et al.*, 2017], derivative calculation [*Roques et al.*, 2017; *Rupp & Selker*, 2006], and fitting method [*X. Chen et al.*, 2018]. To our knowledge, a comparison between these components to find the relative sensitivity is not published. However, we believe this would be a worthwhile venture to determine which part of recession analysis is the most sensitive for parameter estimation.
- 20 We have looked these sensitivities separately. For synthetic recessions, recession parameters are not sensitive to recession extraction methods because w is a constant (see authors reply to major comment 2). Following previous studies, we used common practices for recession extraction criteria. We explored the derivative calculation method's effects on parameter estimation comparing the constant time step method [*Brutsaert & Nieber*, 1977] and the exponential time step method. [*Roques et al.*, 2017]. While we do see artifacts when using the constant time step, parameter estimation between methods are similar
- 25 (see Figure and Table below). In contrast, parameter estimation varies greatly between fitting methods. Because parameter estimation appears to be most sensitive to the fitting method, we have used the citation for previous studies looking at recession extraction and the derivative method and focused our analysis on the sensitivity of the fitting method for parameter estimation.

|    | CTS - a |       | CTS - b |      | ETS - a |       | ETS-b |      |
|----|-----------------------|-------|---------|------|---------|-------|-------|------|
|    | early                 | late  | early   | late | early   | late  | early | late |
| LE | -12.7                 | -6.1  | 2.7     | 0.8  | -11.2   | -5.8  | 2.4   | 0.9  |
| СТ | -7.0                  | -4.1  | 1.9     | 1.0  | -6.9    | -4.2  | 1.9   | 1.0  |
| BA | -6.1                  | -3.7  | 1.8     | 1.1  | -6.1    | -3.9  | 1.8   | 1.2  |
| MI | -12.7                 | -20.3 | 3.2     | 7.6  | -10     | -17.5 | 2.7   | 6.4  |

**Minor comments**

P01L21 Here the Brutsaert and Nieber and the Kirchner paper are missing.

5 Authors Reply: Corrected as suggested

P01L24 Add Brutsaert and Nieber, 1977 as a first fundamental paper using dQ/dt-Q.

Authors Reply: Corrected as suggested

P02L03-07 Better change the two sentences: first collective recession analysis remains common, then the recent criticism.

Authors Reply: Corrected as suggested

10 P02L10 Lower envelope has not been introduced yet.

Authors Reply: Wording clarified- LE introduced P02L08 and wording changed on P02L10

P02L20/21 Explain shorty what a greater b means in terms of recession behavior.

Authors Reply: Done- P02L21-23

P03L11 Last sentence of section 1 is rather a result or a conclusion, but can be rewritten to a hypothesis or research question.

15 Authors Reply: We have rewritten this sentence as a research objective.

P03L14 I would change the order; first the definition of hydrographs then the parameter estimation methods (which needs the hydrographs to be applied).

**Authors Reply: Corrected as suggested**

P03L24 Would be helpful to give key references here for the four fitting methods and to state if the original procedures from the references are used or modified.

Authors Reply: Corrected as suggested. For consistency, references have been added in the methods that were first mentioned

5 in the introduction.

P03L27 Why 5% (and not 10%), is 5% common?

Authors Reply: We have added a citation to this sentence with other authors who have used 5%. While Mendoza et. al. (2003) uses 10% and Vannier et. al. (2014) uses 2%, using a quantile threshold of 5% appears to be the most prevalent in the literature. P06L18 Last paragraph of section 3.2 belongs rather to the methods than to the results section. At least, a reference in the

10 methods section referring to this result should be given.

Authors Reply: Corrected as suggested

P08L21: "sum of the squares", I don't get this.

Authors Reply: Unclear wording removed

**Technical comments**

P01L13 "based on dQ/dt-Q-recession analysis"
Authors Reply: Corrected as suggested
P02L01 why "primary"?
Authors Reply: We have removed this wording.

P02L14: two times Kirchner reference, remove one.

20 Authors Reply: Corrected as suggested

P02L24 "significantly"? sound like it was statistically proven?

Authors Reply: Thank you for this comment. This qualifier has been removed to avoid a misleading statement.

P03L05 "Being able to. . . vulnerability studies" - the sentence is not well connected here, move up?

Authors Reply: We have reworded this sentence and relocated to increase flow.

25 P03L22 "theatrical"?

Authors Reply: Corrected as suggested.

P08L22 to understand the underlying hydrology

Authors Reply: Corrected as suggested.

P08L23 Sentence is a repetition of the last sentence of the paragraph before sentence removed

30 Authors Reply: We have removed this sentence to reduce repetition.

----General Comments------

While I agree with the authors that there is "work yet to be done" in the field of streamflow recession analysis, and really

5 appreciate some of the authors' discussion points on the practical implications of using point cloud analysis vs. individual event analysis (Page 8), there are a number of parts in this manuscript that I find difficult to understand, or that I believe impose "baked in" sources of variability that may not reflect the forms of variability imposed by actual physical processes.

1) In some respects, it seems that Case 1 really encapsulates the main point of this paper (which numerous authors have already argued; though I think the point is worthy of reiteration); that individual recession events typically have steeper slopes than

- 10 the best fit slope of a line through a point cloud generated by a collection of individual events. Case 1 demonstrates how this might happen; if the recession scale parameter (a) scales with the initial flow condition of the recession (Q0), the intercept of the recession curve in log-log space will shift up or down with Q0, and so a collection of steep recession curves will "stack" in a such a way as to create a point cloud that is less steep than the individual curves of which it is composed (see specific comment #5 for additional comment on imposing this form of variability in 'a'). Cases 2 and 3 are primarily used in the
- 15 Discussion and Conclusion to demonstrate this same point. For this reason, I do not see how these cases are useful. These cases might be advantageous if the authors were able to systematically explore the effects of the magnitude-frequency distribution of recharge events on individual recession curves (for example, convincingly attributing the spread in "b" to the magnitude-frequency statistics of recharge). However, I would argue, there is no clear way to perform such a systematic exploration given our present understanding of the physical origins of power law recession dynamics.
- 20 Authors Reply: Thank you for raising this important point. We agree that Case 1 illustrates a key point that individual recessions can have a steeper slope than the best fit line through the point cloud that many authors have already presented. However, individual recessions in Case 1 do not have variability in b values. The example given in the introduction of Lookout Creek (Figure 1) shows a wide distribution of b values, consistent with other studies showing a range of individual b values. Case 2 and 3 are presented with the hypothesis that superposition of flow events changes the value of w which can produce a
- 25 distribution of b values. We agree that a systematic exploration of the physical origins of recession analysis can be valuable and should be explored in the future. While we have not provided a relationship between the variability of b and recharge magnitude and frequency, we believe that the cases presented by controlling for event hydrology are valuable for determining the effects on individual events. We suggest that the event magnitudes distribution produces an offset of individual events while event spacing changes the event length and antecedent flow contribution while antecedent events change the effective
- 30 decay constant and thus the b value for individual recessions.

We have developed a paragraph in the conclusions to explain why each of the three Cases contribute of the findings of this paper and give more equal weight to the contribution of each of the three cases: "For Case 1, the recession analysis parameter *a* is equal to  $w/(\tau Q_o^{1/w})$  and thus the intercept of the individual recession curve will scale with  $Q_o$ . The result is a collection of

individual recession curves that are horizontally offset based on the initial discharge producing a smaller *b* value for the point cloud compared to the individual recessions. Case 1 illustrates that the slope of individual recession events can greater than the best fit line through the point cloud, consistent with previous studies (Biswal and Marani, 2010; Mutzner et al., 2013; Shaw and Riha, 2012). However, the point cloud in Case 1 is generated by a collection of multiple individual recessions all with the

- 5 same slope and does not have the variability in *b* values presented in these same studies and shown for Lookout Creek in Figure 1. Case 2 and 3 are presented using superposition of antecedent flow events that consequentially changes the individual recession *b* values, providing a possible explanation for the variability in *b* values for individual recessions. For Case 2, the superposition of events takes into account antecedent conditions which results in a distribution of individual recession *b* values where *b* values are associated with the baseflow contribution. For Case 3, the horizontal offset of individual recession from
- 10 Case 1 and the effects of antecedent conditions from Case 2 result in the variability of individual recession *b* values that are horizontally offset."

2) The methods need to (a) more thoroughly explain exactly how to generate the various forms of synthetic recession, and (b) how these different forms might reflect the impacts of real, physical processes in a watershed. On the first point:

(a)

- 15 i. How do the authors translate a recharge magnitude (presumably with units of [L]?) into a flow increment (with units of [L/T])? In the case of nonlinear recessions, the flow increment is a nonlinear, flow dependent function of the recharge depth. Authors Reply: Thank you for this interesting comment. We now have included the relationship between recharge and flow in the synthetic hydrograph methods section. A pulse recharge can be calculated by integrating the equation for the falling limb of the hydrograph such that the recharge volume is:  $V = DA = \tau Q_o/(w 1)$  for w > 1, where *D* is depth of recharge over
- 20 area A. Interestingly for w<=1, the recharge is an infinite volume. In the previous version of the manuscript w was 0.7 which would result in infinite recharge. As a result, we have modified the value of w in all examples to be 1.2 representing non-linear reservoir behavior and updated the associated figures.

ii. What are the parameterizations used for the various distributions from which flow increments and inter-arrival times are sampled?

- Authors Reply: Cases 1&3 have log-normally distributed inter-arrival times ( $\mu = 2.5$ ,  $\sigma = 1$ ) and event magnitudes ( $\mu = 1$ ,  $\sigma = 1$ ), compared to Case 2 with uniform event inter-arrival time ( $\mu = 450/\tau$ ), and magnitudes ( $\mu = 1$ ). The time series is based on concatenating successive individual events based on the guidelines for each case (see Table 1). We have developed a paragraph in the methods section where we discuss the distributions of how the synthetic curves were generated.
- iii. How am I to interpret the .mat file uploaded to Hydroshare? I loaded this file, and I see there are columns "mag",
  30 "start\_locs", and "value". How do I use this information to reconstruct the recession curves the authors analyzed? It's undocumented, and not described in the text.

Authors Reply: Thank you for bringing this to our attention. We now have provided a description to the Hydroshare file that includes the file structure information and a subset of Matlab code that identified how the columns were constructed including

the parameters used for the distributions. To increase accessibility, we have also included the information in a CSV in addition to the Matlab files on the Hydroshare site.

iv. For Cases 2 and 3, can the authors more clearly define their superposition procedure? Going off of Figure 2, how is the "underlying second event" (QC) constructed? Is the recharge increment added to the value of flow at the end of the previous

- 5 recession? Or is this how QA is generated? One possibility for QC (once the authors clarify how it is constructed) is that we have effectively created a second "reservoir" with an initial storage equal to the magnitude of the recharge event. Then, QD would equal the sum of the discharge from the continued first event and the discharge from the second reservoir. Authors Reply: Thank you for this comment, we agree that the superposition procedure needed to be better defined. We have
- now better described the superposition procedure used to create Case 2 & 3 following the nomenclature in Figure 2. "For Cases 2 and 3, individual recessions were linearly superimposed on antecedent flows. The baseflow from the first event,  $Q_B$ , is an extrapolation from the first event using a constant power law decay constant. The underlying second event,  $Q_C$ , is defined by the event magnitude given by the random number generator and a defined power law decay constant. The peak flow for  $Q_A$  is based on adding  $Q_B(t_0)$  to  $Q_C(t_0)$  while the decay constant is based on the underlying second even,  $Q_C$ . Case 1 is based on the hydrograph represented in  $Q_A$ . The resulting flow from superposition,  $Q_D$ , defines the peak flow the same as  $Q_A$  but the decay
- constant changes based on the linear superposition of QB and QC. For Cases 2 and 3, QD represents the hydrograph structure."
   (b)

Referring to comment (iv.) above, it's not clear how this appearance of a second reservoir represents any physical process, or why it's a meaningful way to generate variability. The idea that the previous event recession somehow continues unabated and superposed with the current event effectively splits the watershed into two parallel components that, owing simply to the

- 20 occurrence of a recharge event, now operate independently of one another. The procedure amounts to taking the sum of two nonlinear reservoirs with identical values of 'b' (page 5, Line 23), and varying value of 'a' imposed by Page 4, Line 17. I don't disagree that this will generate a new recession curve with entirely different power law parameters which depend on previous flow conditions, but the authors do not provide a rationale for imposing this form of memory. A more defensible approach (in my opinion) taken by previous authors is to explicitly acknowledge physical mechanisms that might give rise to parallel
- 25 reservoirs throughout a landscape (for example, conceptualizing a watershed as a collection of contributing hillslopes with varying hydraulic response times). In such cases, parallel reservoirs may generate increased nonlinearity, as demonstrated by Harman et al (2009) and Gao et al (2017). While it is true that these previous authors use superposition of linear reservoirs, the actual dynamics that give rise to increased nonlinearity are similar to those operating in the present work. On a related note, I think the authors should be citing these previous manuscripts, which I believe are very closely related to the present
- 30 work.

Authors Reply: The reviewer brings up an important question about the physical representation of the independent reservoirs which gives us the opportunity to clarify the motivation for how we combined individual events to create the hydrograph. Appealing to the simplest model presented by the instantaneous unit hydrograph method by Dooge (1973), we utilize the simplest model of linear superposition. We acknowledge that there are other ways to create watershed memory that would also

generate variability, and the effects of parameter estimation from different reservoir models for combing events into a hydrograph would be worthwhile research to peruse. However, for the purposes of this paper providing a linear superposition for the hydrograph shows generated variability between recessions as a simple possible representation. We have included a discussion in the synthetic hydrograph methods section that describes the choice for a simple model based on the instantaneous

5 unit hydrograph and the kinematic wave model. In the discussion and conclusions section, we have included text that introduces the concerns about the linear superposition the reviewer raised and expressed the need for future work on parameter estimation sensitivity on the different reservoir models for synthetic hydrographs.

We agree that citations for reservoir theory relating to recession analysis should be included as it is intimately related to how the synthetic hydrograph is created and interpreted. We have included the citations on reservoir theory to values for w and b

- 10 with regards to the relationship between recharge and flow (see author reply for general comment 2ai for the relationship between recharge and flow). For w>1 based on the recharge equation, a value for b>2 can be achieved by combining discharge from multiple linear reservoirs in parallel, (e.g., Clark et al., 2009; Gao et al., 2017; Harman et al., 2009; Rupp et al., 2009), multiple linear reservoirs in series (e.g., Rupp et al., 2009; Wang, 2011), or multiple nonlinear reservoirs (e.g., McMillan et al., 2011). We have taken this comment into consideration and included reference to b values by using reservoir theory.
- 15 3) I do not understand the purpose of the "early" vs. "late" fitting method in the context of this work. The early/late time methodology derives from the analysis of Brutsaert and Nieber [1977], who show that a shift from a recession slope of 3 to 1.5 is a direct consequence of the dynamics of a Boussinesq-style hillslope groundwater table. The physical implications of the authors choices in construction of synthetic hydrographs (e.g. existence of parallel reservoirs in the previous comment) are not necessarily consistent with the dynamics of a single hillslope groundwater table, so why use a form of analysis that is
- 20 specific to the Boussinesq framework?

Authors Reply: Thank you for this comment. The introduction of early vs time-time was intended to be used for comparison of parameter fitting methods and not to be applied towards the synthetic hydrographs. For clarification, we have elaborated on the choice of using early and late-time for the parameter estimation fitting methods section and not for the synthetic hydrographs. In the parameter estimation fitting methods section, a description has been added to clarify that because a change

- 25 of hydraulic regime was suggestive in Figure 1 between high flow ranges and low flow ranges, recession analysis parameters were estimated for two flow ranges, early-time and late-time. The differentiation of recession analysis parameters into early and late-time was chosen to capture the change in hydraulic regime in Figure 1 and to provide b values for the lower envelope fitting methods. In the text when the LE fit is discussed, we have noted that a limitation of using a pre-defined value for b assumes that the watershed responds like a homogenous Boussinesq hillslope with behavior similar to a single hillslope, which
- 30 isn't known priori for the LE of a watershed composed of multiple heterogeneous hillslopes of unknown b. For the synthetic hydrographs, they do not exhibit a change in hydraulic regime and thus a single fit is used for each fitting method because w is a constant value.

------Specific comments-------

1) Page 2, Lines 16 - 18: Do the authors intend to say that sources of variability in a,b between events may derive from these sources? Also, it is not clear what the authors mean by "flow superposition from previous events".

Authors Reply: We have changed the wording to avoid confusion.

2) Page 2, Line 29: This statement is vague; of course the hydrology of a recession event affects the recession event.

5 Authors Reply: We refocused this statement and made it more aligned with our hypothesis.

3) Page 3, Line 9: Use of superposition without defining the term.

Authors Reply: We have replaced the term in this instance to avoid confusing before the term is defined.

4) Page 3, Line 22: "theoretical"

Corrected as suggested.

- 10 5) Page 4, Line 15 17: While I agree that this is certainly one way to introduce variability in the recession scale parameter, it is nevertheless arbitrary to impose this particular relationship pinned to a 45 day timescale. Subsequent interpretation should be qualfied with "where a = -w/(t0\*Q0^(1/w)) holds. . .". While it is a convenient expression for imposing variability in 'a', I am unaware of any process-oriented result that shows the recession scale parameter should be determined in this way. Related to this, on Page 6, Lines 3 12, this discussion is difficult to follow. I think the authors are making the point that within their
- 15 imposed timescale framework, the recession scale parameter 'a' must collapse to a single value that no longer depends on the flow initial condition in the limit as b=1. I agree that, mathematically, this is what happens, but the authors don't provide a compelling case that this is physically what should happen with real recessions; so the conclusion, "yet this result suggests that a condition where b=1..." should be qualified with the requirement that this would be true in circumstances where the authors' imposed form of variability for 'a' holds.

20 Authors Reply: Corrected as suggested.

6) I assume the authors meant to put "3 Results" not "3 Methods" on page 5.

Authors Reply: Corrected as suggested.

7) Page 7 Line 23: "We hypothesize. . ." is an almost tautological statement.

Authors Reply: We have elaborated on this sentence to include our hypothesis about the variability of individual recession

analysis parameters.

- Figure 2: Why is there a "t0" at the top of the plot? Isn't t0 the 45 day timescale imposed to generate the recession scale parameter?

Authors Reply: Thank you for this comment. We have corrected this unclear notation. Now he characteristic timescale has

30 been changed to the symbol  $\tau$  to avoid confusion. The t0 in Figure 2 signifies the start of the recession, and the equation evaluates the discharge as a time since the recession start.

**Recession analysis 42 years later - work yet to be done**Recession analysis revisited: impacts of climate on parameter estimation**

Elizabeth R. Jachens1, David E. Rupp2, Clément Roques3, John S. Selker1

1Department of Biological and Ecologic Engineering, Oregon State University, Corvallis, OR, 97330, USA
 2Oregon Climate Change Research Institute, College of Earth, Oceanic, and Atmospheric Sciences, Oregon State University, Corvallis, OR, 97330, USA
 3Department of Earth Sciences, ETH Zurich, 8092 Zürich, Switzerland

Correspondence to: Elizabeth R. Jachens (jachense@oregonstate.eduerjachens@gmail.com)

- 10 Abstract. Recession analysis is a classical method-employed in hydrology to assess watersheds' hydrological properties by means of the receding limb of a hydrograph, frequently expressed as the rate of change in discharge (-dQ/dt) against discharge (Q). This relationship is often assumed to take the form of a power law -dQ/dt=aQb where a and b are recession parameters. Recent studies have highlighted major differences in the estimation of the recession parameters depending on the method, casting doubt on our ability to properly evaluate and compare hydrological properties across watersheds based on -dQ/dt vs.
  15 O recession analysis. This study shows that estimation based on collective recessions as an average watershed response is
- 15 *Q* recession analysis. This study shows that estimation based on collective recessions as an average watershed response is strongly affected by the distributions of event inter-arrival time, magnitudes, and antecedent conditions, implying that the resulting recession parameters do not represent watershed properties as much as they represent the climate. The elear-main conclusion outcome from this work highlights is that proper evaluation of watershed properties is only ensured by considering independently using recession analysis requires considering individual recession events. While average properties can be
- 20 assessed by considering the average (or median) values of *a* and *b*, their variabilities provide critical insight into the senstivibility of a watershed to the initial conditions involved prior to each recharge event.

**1** Introduction**

Accurate representations of watershed-scale hydrological processes involved in the critical zone is are urgent in a global and anthropogenic change perspective. Streamflow recession analysis has been routinely used for about half a century

25 to assess-a watershed properties (e.g., Brutsaert and Nieber, 1977; Kirchner, 2009; Mcmillan et al., 2014) and more recently their vulnerability to climatic and anthropogenic factors (Berghuijs et al., 2016; Brooks et al., 2015; Buttle, 2018; Fan et al., 2019). Recession analysis is commonly presented-done by plotting the time rate of change in discharge -dQ/dt as-vs.a function

of discharge Q within a bilogarithmic axesplot.- Theory for an idealized single aquifer predicts a power law relationship with parameters a and b (Brutsaert and Nieber, 1977; Rupp and Selker, 2005).

 $-dQ/dt = aQ^b$ (1)

that can be expressed by the slope *b* and intercept log(a) in log transformed space: i.e. log(dQ/dt) = b log(Q) + log(a)(Brutsaert and Nieber, 1977; Rupp and Selker, 2005)(Brutsaert and Nieber, 1977).

However, it has been long been recognized for more than 15 years that the accuracy in the estimation of those parameters is highly sensitive to the methods used (Chen et al., 2018; Dralle et al., 2017; Roques et al., 2017; Rupp and Selker, 2006a; Santos et al., 2019; Stoelzle et al., 2013).

The two primary categories of parameter estimation methods are based on: 1) the collection-aggregation of all observations in
-dQ/dt vs. Q points, space, hereafter referred to as the "point cloud", to describe as the average watershed behavior-over time; and 2) the identification of using\_individual recession events ins -dQ/dt vs. Q space to look at the variability of a watershed's response (Roques et al., 2017). There is a long history of recession analysis parameter estimation using the point cloud beginning with Brutsaert and Nieber (1977) and it remains common (e.g., Buttle, 2018; Liu et al., 2016; Meriö et al., 2019; Ploum et al., 2019; Sánchez-Murillo et al., 2015; Stewart, 2015; Vannier et al., 2014; Yeh and Huang, 2019). In recent literature

- 15 there has been a shift toward-using away from using collective recessions to evaluate recession parameters in favor of using individual recessions to estimate recession parameters (Basso et al., 2015; Karlsen et al., 2018; Roques et al., 2017) and Santos et al. (2019) go as far as to question the validity of point cloud estimation methods. These latter\_studies have not discredited the use of the point cloud, but rather suggest the individual recessions as a more accurate an alternative. However, the use of collective recessions remains common (Brutsaert, 2008; Ploum et al., 2019; Sánchez Murillo et al., 2015; Stewart, 2015;
- 20 Stoelzle et al., 2013).

5

When Brutseart and Nieber (1977) recession analysis was first proposed their recession analysis method-in 1977, aquifer recession behavior was identified by fitting athe lower envelope toof the point cloud, thus by assuming small values of -dQ/dt for a given Q represent adrought quifer discharge flow and anything larger has quick flow contributions from faster pathways such as overland flow. (Brutsaert & Lopez, 1998; Brutsaert & Nieber, 1977). The Thisis lower-envelope method of estimating

- 25 fitting recession analysis parameters using the lower envelope method was shown to be highly subject to artifacts arising from instrumentmeasurement noise and recording precision (Rupp and Selker, 2006a; Troch et al., 1993), and improvements to fitting a lower envelope-methods have been proposed (Stoelzle et al., 2013; Thomas et al., 2015). An alternative fitting method wherein *b* was estimated as the best linear fit to the point cloud was introduced by Vogel and Kroll (1992) as the central tendency (Vogel and Kroll, 1992)(e.g. Rupp et al., 2009). The central tendency method was adapted by Kirchner (2009) to
- 30 address the undue weight of highly uncertain extreme points. Kirchner (2009) instead-also suggested fitting a polynomial

function to averages within bins of the cloud data-(Kirchner, 2009). All of these point cloud fitting approaches fundamentally treat each computation of dQ/dt and Q as reflecting a single average underlying curve, with the variability deviations from a single curve of individual events effectively treated as noise.- In other studies, dData have been subsetted by season or month (e.g., Szilagyi et al., 2007; Thomas et al., 2015) to examine seasonal variations in the recession characteristics parameters with

5 the subsets still treated to point cloud analyses.

effectively ignoring the variability of individual events. The variability between events may derive from various factors includinginclude but not limited to, initial conditions, event recharge magnitude, duration of the recession event, spatial distribution of the recharge event within the watershed, and the flow superposition from previous eventsthe flow decay rate.

In contrast, tThe variability in watershed response to individual recharge events can be depicted by fitting recession parameters

- 10 to looking specifically at individual recession events. Several aAuthors have observed that individual recessions had greater b than did the point cloud\_(Biswal and Marani, 2010; Mcmillan et al., 2011; 2014; McMillan et al., 2011; Mutzner et al., 2013; Shaw and Riha, 2012); -a larger value of b indicates a time rate of decline- that decreases more quickly with decreasing streamflowslower rate of decline in streamflow decline b value describes the rate of discharge decline, with a greater value indicating more stable streamflow while a smaller value indicates streamflow with a greater streamflow decline rate. Consistent
- 15 with previous literature these studies, we have also observed individual recessions that have a larger *b* than the point cloud fit across watersheds in the Oregon Cascades. To serve A-as an example, we present in Fig. 1a recessions -analysis plot in Figure 4 for the Lookout Creek, Oregon, USA, using -daily discharge data from 1949 to 2016 (station USGS# 14161500) (Johnson and Rothacher, 2019; obtaineed from USGS, 2019). In the 66 years of data presented, a total of 1309 recession events are identified with, an average of 19 events per year. It is clearly evidentstriking to see that values of *b* for individual recession
- 20 events tend to be significantly larger than *b* for the point cloud particularly for those at lower discharges. In this example,  $_{-5}$  individual event selection criteria include recessions lasting longer than 5 days, starting 1 day after the peak to exclude the influence of overland flow, and ending at the next following-precipitation event, following other studies. The minimum event duration of 5 days is consistent with other studies that use daily averaged data (Biswal and Marani, 2010; Shaw and Riha, 2012). The *b* parameter estimated using point cloud analysis (binning average method) is smaller (-*b* = 1.5) compared to
- 25 median of *b* from the individual recessions (individual *b* = 2.8, with 50% of individual recessions taking values from 2.0 to 4.7; (standard deviation = 4.2) (See colorbar of Fig.ure 1). The statistical frequency distribution of the *b* parameter from the the variability of individual recessions seems to is skewed right and roughly lognormal which suggests that *b* from the point cloud does not represent an average or 'master' recession behavior. hydrology of the individual events influence the individual recession curves.
- 30 For a given discharge range, there appear to be multiple individual recessions that are horizontally offset but appear to conserve the parameter *b*, whereas *a* is not conserved. The offset of individual recession events suggests that antecedent conditions may influence recession analysis coefficients and thus the point cloud may only represent the variability of individual recessions

[revised manuscript text omitted]

- 5 a large number of basin when fitting Eq. (12) to with b = 1 to point cloud data The characteristic timescale is assumed to be 45 days and constant between individual recessions (Brutsaert, 2008). Consequentially, *a* is variable and equal to  $-w/(\tau t_0 Q_0^{-1/w})$ . Where  $a = w/(\tau Q_0^{-1/w})$  holds. By maintaining a constant characteristic timescale the result results in is a hysteretic dQ/dt vs. *Q* relationship, in contrast to the constant *a* value which produces a single non-hysteric relationship. It remains to be seen whether such a similarly-a narrow distribution of  $\tau$  occurs for when *b* not equal to is not set equal to 1. Consequentially, *a* is variable and
- 10 equal to  $w/(t_{\Theta}Q_{\Theta}^{1/w})$ .

A pulse recharge amount corresponding to a given  $Q_0$  can be calculated by integrating Eq. (24) from t = 0 to  $t = \infty$ . For w > 1 (b < 2), the recharge volume is

$$V = DA = \tau Q_o / (w - 1)_{-,}$$
(3)

where D is the depth of recharge and A is the aquifer area. For For  $w \le 1$  ( $b \ge 2$ ), integrating Eq. (24) results in an infinite

volume, so b > 2 can only be sustained over a finite part of any recession. VHowever, values of b > 2 have been derived from physical theory for the early portion of a recession (Brutsaert and Nieber, 1977; Rupp and Selker, 2005) or can-still be obtained from recession curves over a finite time period while retainingwith physical realism, -albeit over a finite period to avoid non-physical results, by combining discharge from multiple linear (b = 1) or non-linear (1 < b < 2) reservoirs (e.g., McMillan et al., 2011). The effect on b of combining linear reservoirs in parallel (e.g., Clark et al., 2009; Gao et al., 2017; Harman et al., 2009; Rupp et al., 2009) and series (e.g., Rupp et al., 2009; Wang, 2011) has received much more attention.</li>

We compared three hypothetical cases-time series generated with different assumptions about the distribution of the magnitudes and inter-arrival times of recessionecharge events, and the superposition of recession events with different

hydrologic controls (Table 1). This design allowed us to specifically examine hydrologic controls in order to assess their influences on individual recessions. We can also determine the influence of parameter estimation for individual recessions and
the point cloud. The hydrologic controls we looked at were the inter arrival time and magnitude of recharge events and antecedent conditions controlled by the falling limb *w* control the distribution of individual recession analysis events and thus the point cloud. The inter-arrival times of events are distributed log-normally (Cases 1 and 3) or uniformly (Case 2). Event magnitudes (as defined given by *Q*0) are either distributed log-normally (Cases 1 and 3) or haveall of the constant same

magnitude (Case 2). Events are either independent of antecedent conditions (Case 1); or events are superimposed on antecedent 30 conditions (Cases 2 and & 3) (Table 1 and Fig. 2).

To creategenerate the time series for the three casesCase 1 and 3, independent recessions were created using a random number generator for a lognormal distribution for event peak magnitude and duration for a total of 10 years of time-series data. The lognormal distributions for event magnitude and duration are chosen for the synthetic hydrographs because the distributions for Lookout Creek are skewed right and roughly lognormal which is also consistent with other skewed right precipitation

- 5 distributions in previous studies (Begueria et al., 2009; Selker and Haith, 1990).- For each of the 3 cases, a time series was created by taking the independent recessions and combining based on the antecedent condition. For Case 1, Recharge eEevents were created with log-normally distributed inter-arrival times ( $\mu = 2.5, \sigma = 1$ ) and event magnitudes ( $\mu = 1$  day,  $\sigma = 1$ ) where both values are normalized by timescale and the unit hydrograph respectively. The normal mean of 2.5 for the inter arrival timeThese values of  $\mu$  and  $\sigma$  results in event lengths with a mean of 20 days, and awith an average of 18 events per year. This
- 10 value was chosen to be comparable to the 19 events per year identified in the Lookout Creek eExample. The normal standard deviation of 1 fo The distributions of  $\mathbf{r}$  both the inter-arrival times and event magnitudes result in distributions that are skewed right, representing the high frequency of smaller events and less frequent large events. Changing  $\mu$  and  $\sigma$ the normal mean and standard deviation will modify the amount of variability in individual recessions, and could be further explored with different distributions in future research regarding the resulting variability in *b*. For Case 1, the individual recessions were combined to
- 15 make a time series such that each event was concatenated onto the last event disregarding the antecedent flows. For Case 2, one for Case 2 assumes constantwith uniform event inter-arrival time ( $\mu = 450/\tau$ ) and magnitudes ( $\mu = 1$ ). The mean interarrival time of 10 days is intended to be comparable with the 19 events per year identified in the Lookout Creek eExample.

For Case 1, the individual recessions were combined to make a time series such that each event was concatenated onto the last event disregarding the antecedent flows. For Case 2 and 3, individual recessions were linearly superimposed on antecedent

- 20 flows, Appealingappealing to the simplest model presented by the instantaneous unit hydrograph method (Dooge, 1973). (,), we utilize the simplest model of linear superposition; we discuss the implicit assumptions of this model in the Discussions and Conclusions section. From Fig. 2, the baseflow from the first event,  $Q_B$ , is a simplen continuation of extrapolation the first recession from the first event using a constant power law decay constant. The underlying second event,  $Q_C$ , is defined by the second event's initial magnitude (constant in Case 2 and given brandomly generated y the random number generator in Case
- 25 3)and a defined power law decay constant. The peak flow for QA is based on adding QB(t0) to QC(t0) while the decay constant is based on the underlying second even, QC. Case 1 is based on the hydrograph represented in QA. The resulting flow-from superposition, QD, is the sum of QB and QC, defines the peak flow the same as QA but the decay constant changes based on the linear superposition of QB and QC. For Cases 2 and 3, QD represents the hydrograph structure. For Case 3, the inter arrival time and event magnitudes are equal to Case 1 while the procedure for combining recessions is equal to Case 2.
- 30 As a result, Case 1 looks specifically at a time series events where the falling limb of each event maintains the same decay constant, and the effects of having no antecedent baseflow influence on streamflow. By including baseflow to Case 2 but maintaining equal inter-arrival times and event magnitudes, we look specifically at the effect of antecedent conditions on

individual recessions and the point cloud. Case 3 combines the distribution of event inter-arrival times and magnitudes of Case 1 with the baseflow conditions of Case 2, best representing the variability and inter-arrival times of individual recession events seen in Fig27 1 for data from Lookout Creek. Each case will address how the controls on the hydrograph affect the recession analysis plot and the -estimatesd of by a and b.

**5 [Insert Fig. 2]**

10

**[Insert Table 1]**

After defining the hydrograph, recession extraction and parameter estimation were performed. Because events are based on a synthetic hydrograph, recession extraction was based on the individually defined events that make up the hydrograph. The beginning of the recession was defined as the peak in discharge because no potential influence of overland flows exist. Events of any length were included with the end of the recession defined as the time step before the next chronological peak. The exponential time step method was used for derivative calculation.

**2.2 Recession Parameter Extraction Methods**

From the observed hydrograph, identification of the recession period is necessary for recession analysis. Recession extraction from observed the hydrographs and the associated sensitivities to different criteria has been summarized explored by Dralle et

15 al. (2017), including minimum recession length, and the the definition of the beginning and the end of the event. For Lookout Creek, wWe used extraction criteria similar to those of other studies (e.g., Chen and Krajewski, 2016; Dralle et al., 2017; Stoelzle et al., 2013) and applied We apply In order to address the acknowledged that recession estimation is sensitive to parameter extraction, the same recession extraction guidelines are applied criteria prior to all fitting methods presented in Section 2.3. to By using consistent recession extraction criteria, all fitting methods are utilizing the same recessions thus isolateing differences in calculated *b* values due to fitting method to differences in fitting method-only.2

We used criteria similar to those of other studies (e.g., Chen and Krajewski, 2016; Dralle et al., 2017; Stoelzle et al., 2013) (e.g., Chen and Krajewski, 2016; Dralle et al., 2017; Stoelzle et al., 2013). AAIn this study, When considering real data of Lookout Creek for parameter estimation fitting results (Section 3.1), an individual recession event selection duration must be longer than 5 days. Rainfall data can be used to identify non-interrupted recessions, but rainfall data we will not available in

25 all cases, Without rainfall data available for all watersheds that can be used as a guide for defining periods absent of rainfall, we suggestso we rely on the hydrograph only. The that the start of the recession is defined as one day after the discharge peak to account for the presence of overland flow. (Roques et al., 2017). The end of the recession occurs at the minimum discharge prior to an increase in discharge greater than the errors associated with instrument precision for stage height of ~0.01 ft, which translates into errors in discharge from ~0.01-0.1 m3/s depending on the rating curve and the discharge level (Thomas et al.,

30 2015).

To additionally minimize errors associated with recession extraction criteriaFor the , synthetic hydrographs are used in Section 3.2-because individual events are inherently known, thus reducing subjectivity associated with defining each of the three recession extraction cri, teria, eEvents of any length were included-, the recession start was selected at peak discharge because overland flow was not a factor, and with the end of the recession was chosen defined as the time immediately step before the

5 next generated discharge chronological peak-Without the presence of overland flow, no lag after the peak discharge is defined as the recession start (Roques et al., 2017).

**2.31 Parameter Estimation Fitting Methods**

To compare parameter estimation methods and the dependency on the fitting method, first the definition of the transition from early to late time was defined. In an attempt to reduce the subjectivity of distinguishing late time from early time, the breakpoint in discharge separating early from late time behavior was optimized to best represent the analytical solutions. By separating the data into two subgroups, either smaller or larger than a defined breakpoint discharge, the best fit line was determined for each subgroup. The location of the breakpoint is defined so the error between the observed ratio of *b* for the two subgroups and the theatrical ratio (*b*=3 for early and 1.5 for late give a ratio of 2) is minimized, theoretically defining the subgroup above the breakpoint as early time and the subgroup below the breakpoint at late time.

15 Four methods of estimating representative recession In order to compare recession analysis parameters between methods, four fitting methods were evaluated: lower envelope (LE), central tendency method-(CT), binning average\_-(BA)\_(BA; Kirchner, 2009), and the median of individual recessions (MI)\_(Roques et al., 2017). \_Recession parameters for all methods were determined by Llinear\_regression in fitting in-bi-logarithmic space was used with each method for consistency across methods.

Because a change of hydraulic regime was suggestive in Fig.ure 1 between high flow ranges and low flow ranges, recession

- 20 analysis parameters were estimated for two flow ranges, early-time and late-time. Early-time and late-time describe a theoretical transition of flow regimes between high-flow and low-flow ranges (Brutsaert and Nieber, 1977). In an attempt to To reduce the subjectivity of distinguishing between high and low flows, athe breakpoint in discharge separating high from low flow behavior was optimized to best represent the analytical solutions. By separating the data into two subgroups, either smaller or larger than a defined breakpoint discharge, the best fit line was determined for each subgroup. The location of the breakpoint
- 25 is defined so the error between the observed ratio of *b* for the two subgroups and the theoretical ratio (*b*=3 for early and 1.5 for late give a ratio of 2) is minimized, theoretically defining the subgroup above the breakpoint as early-time and the subgroup below the breakpoint at late-time.

-For each of the four estimated methods-evaluated, parameters -estimation was were estimated estimated for the early-time and behavior at high flow ranges and for late-time behavior separately at lower flow ranges. For the LE method, a defined b was

30 fixed to of-3 and 1.5 for early and late-time, respectively, following methodology from-Brutsaert and Nieber (1977) and a was

chosen such where *a* is fit\_using quantile regression such optimizing *a* so that 5% of points were are left below the lower envelope (Brutsaert, 2008; Troch et al., 1993; Wang, 2011).-It should be noted that a limitation of using these a pre-defined values for *b* assumes that the groundwater discharge behaves watershed responds like discharge from a single, initiallysaturated, and homogenous Boussinesq aquifer hillslope with behavior similar to a single hillslope, which isn't known priori

- 5 for the LE of a watershed composed of multiple heterogeneous hillslopes of unknown b. -An alternative method to fitting the lower envelope without a pre-defined value of b was introduced by Thomas et al. (2015) using quantile regression for vto estimate alues for both a and b, but was not usedeonsidered in this study. For the CT method, the fit included all -dQ/dt vs Q points unweighted (Vogel and Kroll, 1992). For the BA method, bins spanned at least 1% of the logarithmic range, and a lineear fit, instead of the polynomial suggested by Kirchner (2009), -was fit to applied to the binneds based on the inverse-
- 10 variance weightingmodified from Kirchner (2009 data.). We dispensed with the inverse-variance bin weighting used by Kirchner (2009) to account for data noise when we applied the method to the synthetic recessions because some bins contained few points with very low variance and therefore were weighted excessivelyDue to the nature of the synthetic hydrograph. Fthe omission of noise in the data results in discrete individual recessions. Bins from BA often contained just a few points with very low variance and thus an infinite weight. This would rarely be the case using real data. Consequentially, we do not
- 15 use inverse variance weighting for the synthetic cases to avoid outlier bins with low variance. Instead, a direct linear fit on the log bins without weights was performed, which is not suggested to be applied to real datasets.

For the MI method, parameters were estimated ion for individual recessions was performed using methodology from following Roques et al. (2017) and the medians of -for *a* and *b* value-were determined-calculated independently from all individual recessions using methodology from Roques et al. (2017). In all cases, the time derivative -dQ/dt was computed using the

20 Exponential Time Step method (ETS) proposed by Roques et al. (2017).

**3 Methods Results**

**3.1 Parameter Estimation for Observed Recessions (Lookout Creek)Fitting Results**

In Fig. 3 we display the recession plot stacking all individual recession resulting in the formation of the point cloud. The different fitting strategies revealed that Tthe LE, CT, and BA methods all fit to the point cloud and all-result in different

- 25 estimations of values *a* and *b* evident when applied to the observed daily averaged streamflow for Lookout Creek: Notably, with\_Eearly-time values of for *a* and *b* resulting in estimates that are over 50% larger for LE (fixed at 1.5) than CT and BA, and late-time values of *b* are 50% and 25% larger for LE than CT and BA, respectively (Fig. 3 and Table 2). Furthermore, parameter estimation is sensitive to the fitting method using the point cloud or individual recessions, notably for the late-time *b* value which is used for climate sensitivity analysis where *b* for MI is 6x greater for the estimation compared to CT (Table *b* value which is used for climate sensitivity analysis where *b* for MI is 6x greater for the estimation compared to CT (Table *b* value which is used for climate sensitivity analysis where *b* for MI is 6x greater for the estimation compared to CT (Table *b* value which is used for climate sensitivity analysis where *b* for MI is 6x greater for the estimation compared to CT (Table *b* value which is used for climate sensitivity analysis where *b* for MI is 6x greater for the estimation compared to CT (Table *b* value which is used for climate sensitivity analysis where *b* for MI is 6x greater for the estimation compared to CT (Table *b* value which is used for climate sensitivity analysis where *b* for MI is 6x greater for the estimation compared to CT (Table *b* value which is used for climate sensitivity analysis where *b* for MI is 6x greater for the estimation compared to CT (Table *b* value sensitivity analysis where *b* for MI is 6x greater for the estimation compared to CT (Table for MI is 6x greater for the estimation compared to CT (Table for MI is 6x greater for the estimation compared to CT (Table for MI is 6x greater for the estimate for MI
- 30 2). The CT and BA methods are fairly consistent with each other for both early and late-time, with BA resulting in smaller a

values tha whereas  $\overline{\text{CT. t}}$  the pre-defined theoretical *b* values for the LE appear to provide poorer fits tofor the point cloud consistent with previous studies that have shown errors associated with the LE method (Rupp and Selker, 2006a).

More importantly, parameter estimation differs greatly whether the point cloud or individual recessions are used. The late-time *b* value which defines the low-flow baseflow regime is 6 times greater for MI than CT (Table 2). Using the MI method, the *b*

5 value is larger than any other method for both early and late\_-time. Hereafter and for the synthetic hydrographs, we use the binning average method (BA) and the median individual recessions (MI) to compare between the point cloud and individual recessions, respectively, for parameter estimation for the synthetic hydrographs presented.

[Insert Fig. 3]

[Insert Table 2]

**10 **3.2 Synthetic Hydrograph Results**

Based on the similar results from BA and CTL methods discussed above, and the questionable practice of setting an early- and late-time *b* a priori as we did in the LE method, hereafter we use the BA method to represent to point cloud recession parameter estimation when comparing to the MI method using individual recessions.

The recession decay exponent w in Eq. (2) was set to 1.2; distinct values of w were not used for early and late time. This value

- 15 for w results in The falling limb recession for all three cases is defined using w = 0.71.2 as the dimensionless decay constant related to b by w=1/(b-1). The expected value of b =is 2.41.8 for an individual synthetic recession, which -given this decay constant, is which falls within near the range of reported the median of individual b values of 2.0 by-in Biswal and Marani (2010), and 2.1 by bothin Shaw and Riha (2012) and Roques et al. (2017), though less than and the median individual b of 2.8 for Lookout Creek in Figure 1. By using a constant w value, the synthetic hydrographs do not exhibit a change in hydraulie
- 20 regime and thus a single fit is used for each fitting method.

The *b* values and the offset of individual recessions resulting from Eq. (1) are both functions of highly sensitive to the magnitude of the decay constant chosen, *w*. A larger *b* value indicates a more stable baseflow water-discharge (a slower decline rate for given discharge), compared to a smaller *b* value which corresponds to a greater discharge decline rate with decreasing discharge. For a given value of  $b_{\overline{z}}$  and  $\tau$ , a larger value of *a* implievaries inversely with  $Q_0^{1/(b-1)}$ . s a more conductive/permeable

25 basin and a smaller terminal streamflow if the recession continued infinitely. Decreasing  $w_{-}$  would results in larger values of b while also increasing the offset between individual recessions, resulting in a larger range of a values and a more scattered point cloud. In contrast, as w approaches infinity, the offset is minimized as b goes to 1, representing in an exponential falling limb recession in time (Rupp and Woods, 2008). In this special case, the recession analysis plotall of the individual recessions overlap all plot on the same b=1 with constant a (i.e., there is no offset among individual recessions lines). While

b=1 is interpreted as a linear reservoir according to traditional theory and is a convenience often assumed, yet this result suggests that a condition where b=1 and a is a constant given a constant characteristic timescale would not be consistent with the existence of an point cloud, except to the degree at which observation error introduces noise into the recession hydrograph, or other pathways (e.g., overland flow) contribute to the flow in the stream. In summary, the more linear the

- 5 response is (the closer *b* is to 1), the smaller the offset, whereas the more non-linear the response (the larger the *b*), the greater the offset will be and thus the more different the parameter estimations will be between the point cloud and individual recession methods the worse the parameter estimation from the point cloud will be. The 3 subsequent following cases using synthetic hydrographs are intended to highlight the offset of the individual recession curves, using an underlying decay constant of w=0.71.2 that is sensitive enough to showcase the offset of individual recessions but still providing a reasonable recession
- 10 analysis plot. Case 1 uses events with a constant *w* across the hydrograph, while Case 2 & 3 to include superposition of an underlying event with a constant *w* and the antecedent flows which results in a less negative effective decay constant resulting in an increased *b*.

Due to the nature of the synthetic hydrograph, the omission of noise in the data results in discrete individual recessions. Bins from BA often contained just a few points with very low variance and thus an infinite weight. This would rarely be the case using real data. Consequentially, we do not use inverse variance weighting for the synthetic cases to avoid outlier bins with

15 using real data. Consequentially, we do not use inverse variance weighting for the synthetic cases to avoid outlier bins with low variance. Instead, a direct linear fit on the log bins without weights was performed, which is not suggested to be applied to real datasets.

**3.2.1 Case 1**

- Recession analysis of a hydrograph with log-normally distributed event inter-arrival times and peak discharge with a constant
  falling limb decay constant (no baseflow represented) results in individual recession events with the same *b*, horizontally shifted based on the initial discharge (Figure 4). For this case, the peak flow of the event is the only sourcecause of variability in the recession parameter *a*. The variable event magnitudes result in individual events located over a range of ln(*Q*) values whereas - Of interest, if the same flow magnitude was preserved for each event, each individual recession would plot on top of one another creating a single line without a point cloud. The variable event line simple hydrograph representation, parallel individual recessions are present all with *b* = 1.8, as expected. The value of and *b* of the point cloud (is estimated at 1.3) considering the point cloud, which appears to be significantly is significantly much less than imposed the median-individual *b* value of --(1.8). Furthermore, the median individual *b* is equal to the expected *b* (2.41.8) because each recession *t* has the
- same shape. This underestimation results from the offset between individual recessions based on the range of initial discharges
   30 [...].By only considering an event without antecedent conditions, individual recessions maintain a value of b while a is variable. Each individual recession has a b value of 1.8 while the point cloud results in a b value of 1.3 (See colorbar of Figure
  - <del>4).</del>

Regarding To examine the sensitivity of parameter estimation to recession extraction criteria-sensitivity, we evaluated how choosing the start of the recession (i.e., the time elapsed since peak discharge) affects overland flow duration changed the value of *a* when using the point cloud method. Whether we chose 0, 1, or 2 days following peak discharge, *and-a* from -the point cloud *a* wasis 0.17 [-UNITS?] and *b* was 1.3 [-].

[Insert Fig. 4]

5

**3.2.1 Case 2**

The addition of superposition of recession events accounts for the effects of antecedent baseflow. The superposition of the events changes the effective w of the falling limb of the hydrograph as the event recession is added to the antecedent events,

- 10 resulting in variable *b* across the individual recessions in the recession analysis plot (Fig\_ure 5). The median *b* represented is 3.25 -with a the range of individual *b* between 2.56 to and 3.41 (quantile range represented in the colorbar of Fig.ure 5). The point cloud *b* of 2.35 falls outside of the range of *b* values for individual recessions. Superposition results in a larger *b* than what would arise from non-superposition. Steeper recessions (higher *b*) are associated with events with higher baseflow contribution given the same addition of flow. By including antecedent flow conditions, neither *a* nor *b* is preserved between
- 15 individual recessions.

[Insert Fig. 5]

**3.2.1 Case 3**

A hydrograph more representative of real-world-case conditions includes variable inter-arrival times and event magnitudes from Case 1 and baseflow antecedent conditions from Case 2 (Fig. 6a). These complexities result in a recession plot where the individual recessions represent the variability in watershed response represented by the hydrograph (Fig. ure 6b), where *a* and *b* are different between individual recessions. As with Case 1 and 2, the median individual *b* (3.3) is greater than the point cloud *b*\_-(2.0) The minimum individual *b* is 1.9 with a maximum of 8.5 while the point cloud *b* isof 2.0 near the low end of the falls at the lowest range of individual *b* values (See colorbar of Fig. ure 6). The similarity of features of Fig. 6 and Fig. 1 are noteworthy. Though many of the observed recessions in Fig. 1 are slightly curvilinear (in the log-log space) whereas the

25 synthetic recessions are power lawsare linear, in both cases there is a tendency for recessions with lower initial discharges to have higher values of b yet still many instances of recessions with similar initial discharges but different values of b.

[Insert Fig. 6]

**4 Discussion and Conclusions**

20

In the 42 years since Brutsaert and Nieber (1977) proposed their recession analysis, it has provided a seemingly simple analytical method for estimating basin-scale hydrologic properties. However, recent studies have highlighted the sensitivity toof parameter estimation to and the fitting methods on the recession parameter values and to the resultingused and with the

- 5 influence on the interpretation for of average watershed behavior. This paper explores the effect of the distribution (in time and in magntitude) of individual recessions on parameter estimation and compares that to the parameter estimation from for collective recessions (i.e., the point cloud). The four fittingestimation methods considered were the lower envelope method, central tendency, binning, and individual recessions method. Because of the poorer apparent fit and problems pointed out from previous studies when using the , the lower envelope and central tendency methods, we chose to use the binning method to
- 10 compare with results from the individual recessions method for a set of synthetic case studies. were not considered in favor of improved methods for binning of collective recessions and the median individual recession.

We hypothesize that the underlying hydrology climate controls the distribution of individual recessions in bilogarithmic plots of -dQ/dt vs. Q. This distribution which can be related to the variability in recession analysis parameters. Using the three synthetic case studies of synthetic hydrographs, we compare examine the effects of event inter-arrival time, magnitude, and

15 antecedent conditions on the distribution of individual recession events that together comprise the pattern of collective recessions (i.e., the point cloud).

We conclude that recession analysis performed on collective recessions does not capture average watershed behavior, regardless of the fitting method-used with the collective recessions. The frequency between events creates different event lengths that span different ranges of Q. The point cloud is an artifact of the variability of the individual recessions, including the event inter-arrival times and distribution of magnitudes. Individual recessions with the same b but different a can be produced by the same falling limb of the hydrograph at different varying the ranges of ii nitial discharges (Case 1), variability of b for individual recessions can be produced by superimposing events on antecedent flow conditions (Case 2), and different

- recession event lengths with different *b*'s can be produced by including variable event inter-arrival times and magnitudes (Case 3).
- 25 For Case 1, the recession analysis parameter *a* is equal to *w/(τQ01/w)* and thus the intercept of the individual recession curves will scale with *Q0*. The result is a collection of individual recession curves that are horizontally offset based on the initial discharge producing a smaller *b* value for the point cloud compared to the individual recessions. Case 1 illustrates that the slope of individual recession events can be greater than the best fit line through the point cloud, consistent with previous studies (Biswal and Marani, 2010; Mutzner et al., 2013; Shaw and Riha, 2012). However, the point cloud in Case 1 is generated by a collection of multiple individual recessions all with the same slope and does not have the variability in *b* values presented in these same previous studies and shown for Lookout Creek in Fig.ure 1. Case 2 and 3 are presented useing superposition of

antecedent flow events that consequentially changes the individual recession b values, providing a possible explanation for the variability in b values for individual recessions. For Case 2, the superposition of events takes into account of antecedent conditions which results in a distribution of individual recession b values despite the decay exponent w being fixed. where b values are associated with the baseflow contribution. For Case 3, the horizontal offset of individual recessions from Case 1

5 and the effects of antecedent conditions from Case 2 result in the recessions with variabileity of individual-recession b values and that that are horizontally offset to create a pattern similar to that observed in a real watershed.

While the mean *b* for individual recessions in Case 1 is are a direct consequence of the value of *w* used in Eq. (2), representative of the underlying events, this hat is not true when the underlying discharge from each application of Eq. (2), which we call an

- 10 'event', recession-isare superimposed on the antecedent flowflows, as in Case 2 & 3. This superposition of events results in a range of individual recession b2s as often observed in the literature (Basso et al., 2015; Biswal and Marani, 2010; Mcmillan et al., 2014; Mutzner et al., 2013; Shaw and Riha, 2012), thus it appears that the simple superposition of events can recreate the watershed behavior.- However, there is a key underlying assumption to this superposition that is inconsistent with a real watershed. We can consider a spectrum of conceptual models of watershed. On one end of the spectrum is a single bucket with
- 15 a hole near the bottom. The bucket contains porous medium whose properties may vary with depth to create a variety of nonlinear outflows. Each new recharge event adds to the pre-existing storage of water in the bucket. On the other end of the spectrum is what we have done here: Each new event adds water a new and independent bucket and the outflows from all buckets are aggregated. Both conceptual models are patently unrealistic but, remarkably, the latter model produces a distribution of recession events in -dQ/dt vs. Q space that is more like what is observed.- This finding should reveal something
- 20 about the plumbing of the basin.

A first approximation could be using linear superposition of individual events on the antecedent baseflow to back out an underlying recession curve. This underlying recession curve would be a master recession that describes the watershed's underlying hydrology controlling the falling limb of the hydrograph in order to predict watershed recession behavior based on streamflow. The linear superposition of recessions over time effectively adds the outflow of multiple, independent, non-linear

- 25 reservoirs. This adds an infinite number of reservoirs together, where the earlier recessions effectively become negligible after some time. We acknowledge that there are other ways to create memory for the watershed that would also generate variability, including the combination of different linear or non-linear reservoirs (Clark et al., 2009; Gao et al., 2017b; Harman et al., 2009; MeMillan et al., 2011; Rupp et al., 2009; Wang, 2011), and the effects of parameter estimation from different reservoir models for combing events into a synthetic hydrograph would be worthwhile research to peruse. For the purposes of this paper
- 30 providing a linear superposition for the hydrograph shows generated variability between recessions as a result of the inter- arrival time and event intensity instead of geologic controls. We do not mean to imply that linear superposition of individual events fully represents natural watersheds, but instead to present a method that produces variability in *b* similar to what we see

in real watersheds. While using linear superposition of individual events is presented as a simple possible representation that does produce similar results to real watersheds, different representations that describe physical processes of combining individual events appears to be fruitful for more investigation. An additional important simplifying limitation assumption of this study is the use of a constant timescale defined by keeping- $\tau$  constant for each individual event, assuming that the

- 5 underlying events are governed by a single timescale independent of the superposition of events. Previous studies that include have examined analysis of timescales across basins by setting have used b = 1a linear fit and estimating  $\tau$  from the point cloud (Brutsaert, 2008; Lyon et al., 2015). However, given the findinthe gsquestionable of -of this study regarding the validity of the point cloud estimation methods, additional studies -on timescale using individual recessions of would be worthwhile to better understand the variability of timescale among between individual recessions 
[revised manuscript text omitted]

---

## Author Response (AR2)

We appreciate the comment from the reviewer and recognize that there is room to explore other, more complex, representations of watershed response. However, we feel

5 that use of a simple superposition still illustrates our key message without invoking other models. In the future, we look forward to a series of work in this direction from us and others.

We would like to thank the anonymous Reviewer #2 for reviewing our paper for the second round of revisions. Below is a summary of the changes made to the manuscript in response to the comments raised:

- 1. Regarding the appropriateness of applying the instantaneous unit hydrograph (IUH), we have added further clarification that we are using this method as a simple model and acknowledge that we are applying it to non-linear reservoirs and, as such, a key simplifying property of superposition of linear reservoirs as described in Dooge (1973) is not achieved.
- 2. We have expanded the discussion about the physical realism of Cases 2 and 3 to acknowledge that there are other ways to create watershed memory and comparing our results to different models would be worthwhile further research.
- 20 Please find the responses to the comments from Reviewer #2 below, followed by the revised manuscript with tracked-changes. In addition to changes of the manuscript described below in the response to the reviewer and a few minor grammar changes, there has been a change to the units in Table 2 that we wanted to bring to your attention. The units have been modified to be expressed in units of meters of length and seconds for
- 25 time. We look forward to your feedback and the hopeful publication of this manuscript in HESS.

1

Sincerely,

Elizabeth R. Jachens

30

**Author Response to Reviewer 2**

SECOND REVIEW of the paper now titled, "Recession analysis revisited: impacts of climate on parameter estimation" (Jachens et al.)

-----Overview

10

5 I appreciate the new analyses and additional explanations the authors have included in this revised manuscript. Many of my previous issues with the manuscript still stand, especially regarding the physicality of Cases 2 and 3. However, I won't revisit them here.

I have one general comment regarding Cases 2 and 3. The authors appeal to Dooge [1973] and subsequent work on the IUH. Maybe I need to refamiliarize myself with that theory, but I don't think an IUH exists for a nonlinear reservoir; so, this doesn't seem like an appropriate justification for the approach.

Author Response: We recognize this concern and have included additional clarification about our choice to use the IUH. We have added the following sentence it in the text when the IUH is introduced on page 6: "We acknowledge that framework for the instantaneous unit hydrograph does not consider non-linear reservoirs, but use it as a simple representation to produce variability between recessions." We are appealing to the

- 15 superposition of events to create a hydrograph that is represented with the IUH as a mechanism that generates variability between recessions, this is not to say that we are using the IUH directly as described by Dooge [1973]. The implications of this are further discussed in the discussion and conclusion section where we explicitly state that this representation lacks physical realism. We also acknowledge that the uncomplicated approach we considered is not the only, nor necessarily the most appropriate, way to generate watershed
- 20 memory in order to introduce variability. There is undoubtedly more work to be done on the modality of sequencing events for a hydrograph and presumably a more supplicated model might capture different or more nuanced results that the ones presented here.

**-----Comments**

Page 11, Line 20: I would say that the first example is maybe less "patently unrealistic" than the second in that there are definite physical analogues that might be compared to hillslopes, and perhaps constructed in the lab.

Author Response: We agree that the first example would be much less patently unrealistic when the geometry and boundary conditions of a Boussinesq-type aquifer are reasonably well-approximated (such as a single hillslope or lab experiment) though here we mean to consider more complex watershed composed of multiple hillslopes, such as the Lookout Creek basin. We have reworded the text to say that both models have aspects that are not realistic when applied to natural watersheds.

Page 11, Line 20: The authors have described a spectrum with two end-members: one is a single bucket with nonlinear drainage properties, the other is a watershed where the number of buckets equals the number of recharge events. I don't know how these represent two extreme members along some continuous axis of model complexity. As a simple example, where would a watershed modeled as two, parallel linear reservoirs fit on this enertrum?

35 spectrum?

30

Author Response: Maybe the analogy of a spectrum is not very good. However, if the spectrum were not a linear construct, but a "space" of more than one dimension, I suppose one could find a place for two, parallel linear reservoirs along one dimension. However, to avoid such digression in the paper, we have removed the reference to a spectrum and just present these models as two distinctly different conceptualizations.

5 Page 11, Line 21: This is a bit of a strawman comparison; of course the latter produces results more like what is observed because the latter of your two end-members is the only one that can produce variability in recession parameters. Perhaps a more "fair" competition would be to compare the author's Case3 to a parallel reservoirs model, which would allow variation in recession parameters.

Author Response: While it may be a bit of a strawman comparison, we think it is worthwhile because the simple
 non-linear bucket model is what is assumed, at least implicitly, in a large number of recession analyses, beyond just the examples we have cited here. We do agree it would be also worthwhile effort to compare the models brought forward in this study to other established models, which could include two or more parallel linear or even non-linear reservoirs. In the text, we give these as possible avenues of investigation, but the implementation is outside the scope of this paper.

[revised manuscript text omitted]